# Bias in Motion: Theoretical Insights into the Dynamics of Bias in SGD Training

**Anchit Jain**[*]
University of Cambridge
aj625@cantab.ac.uk

**Rozhin Nobahari**
MILA - Quebec AI institute
Université de Montréal

**Aristide Baratin**
Samsung - SAIT AI Lab Montreal

**Stefano Sarao Mannelli**
Gatsby & SWC - University College London
s.saraomannelli@ucl.ac.uk

## Abstract

Machine learning systems often acquire biases by leveraging undesired features in the data, impacting accuracy variably across different sub-populations. Current understanding of bias formation mostly focuses on the initial and final stages of learning, leaving a gap in knowledge regarding the transient dynamics. To address this gap, this paper explores the evolution of bias in a teacher-student setup modeling different data sub-populations with a Gaussian-mixture model. We provide an analytical description of the stochastic gradient descent dynamics of a linear classifier in this setting, which we prove to be exact in high dimension. Notably, our analysis reveals how different properties of sub-populations influence bias at different timescales, showing a shifting preference of the classifier during training. Applying our findings to fairness and robustness, we delineate how and when heterogeneous data and spurious features can generate and amplify bias. We empirically validate our results in more complex scenarios by training deeper networks on synthetic and real datasets, including CIFAR10, MNIST, and CelebA.

## 1 Introduction

Over the past decade, the problem of assessing the fairness of classifiers has garnered significant attention, revealing that machine learning (ML) systems not only reproduce existing biases in the data but also tend to amplify them [21, 40, 11]. Given the complexity of the ML pipeline, isolating and characterising the key drivers of this amplification is challenging. Recent studies have begun to disentangle the contributions from architectural design choices, including overparameterisation [37], model complexity, activation functions [5, 12], learning protocols [43, 13], post-processing practices such as pruning [19], and intrinsic aspects of the data like its geometrical properties [38].

Theoretical results in this area (e.g., [37, 38]) are mostly based on asymptotic analysis, leaving the transient learning regime poorly understood. Due to limitations on computational resources, a trained ML system may operate far from the asymptotic regime and hence existing results may not always apply. Insights from class imbalance literature [43, 12] indicate that classifiers converge faster for classes with more data, but how this applies to fairness, where datasets might be balanced by label but imbalanced by demographics, remains unclear.

Our analysis addresses this gap by providing a precise characterisation of the transient dynamics of online stochastic gradient descent (SGD) in a high dimensional prototypical model of linear

---

[*]Work done as an intern at the Gatsby Computational Neuroscience Unit, University College London

38th Conference on Neural Information Processing Systems (NeurIPS 2024).

classification. We use the teacher-mixture (TM) framework [38], where different data sub-populations are modeled with a mixture of Gaussians, each having its own linear rule (teacher) for determining the labels. Adjusting the parameters of the data distribution in our framework connects models of fairness and spurious correlations, providing a unifying framework and a general set of results applicable to both domains. Remarkably, our study reveals a rich behaviour divided into three learning phases, where different features of data bias the classifier and causing significant deviations from asymptotic predictions. We reproduce our theoretical findings through numerical experiments in more complex settings, demonstrating validity beyond the simplicity of our model.

Our key contributions are:

- **High-dimensional analysis**: We demonstrate that in the high-dimensional limit, relevant properties of the classifier, such as the generalisation error, can be expressed using a few sufficient statistics. We prove that their evolution converges to a set of ordinary differential equations (ODEs) that can be solved explicitly in our setting.

- **Bias evolution characterisation**: Using our solution, we characterise the evolution of bias throughout training, showing a three-phase learning process where bias exhibits non-monotonic behaviour. Specifically:
    1. **Initial phase**: The classifier is initially influenced by sub-populations with strong class imbalance.
    2. **Intermediate phase**: The dynamics shifts towards the saliency, or norm, of the samples in a sub-population.
    3. **Final phase**: Sub-population imbalance, or relative representation, becomes the dominant factor.

- **Empirical validation**: We validate and extend our theoretical results through numerical experiments in both synthetic and real datasets, including CIFAR10, MNIST, and CelebA.

Altogether, our study reveals a complex time-dependence of learning with structured data that previous theoretical studies have failed to capture. This characterisation is crucial for developing effective bias mitigation strategies, especially under limited computational resources.

## 1.1 Further related works

**Class imbalance and fairness.** A key element in our study is the presence of heterogeneous data distributions within the dataset. In the context of fairness, these distributions model different groups in a population. Sampling unbalance is particularly critical, as minority groups are often misclassified [9, 20]. However, theoretical studies on group imbalance have been limited to asymptotic analyses [38], which may not apply in practical settings. Related questions have been explored in the label imbalance literature [22], where it has long been known [1, 17] that underrepresented classes have slower convergence rate and may even experience increased errors early in training. Our work shows that pre-asymptotic analysis can reveal complex transient dynamics, which is practically relevant when learning slows down or training to convergence is not possible. Similar to our analysis, [12] has shown that supposedly neutral choices, like activation functions or pooling operations, can generate strong biases. In contrast to prior work, our focus on data properties identifies several timescales associated to different data features relevant to bias generation.

**Simplicity bias.** Several studies [31, 16, 41, 10, 32] have highlighted a bias of deep neural networks (DNNs) towards *simple* solutions, suggesting this bias is a key to their generalisation performance. Simplicity bias also influences learning dynamics: [4, 32, 28, 30, 33] have showed that DNNs learn progressively more complex functions during training, with a notion of complexity often defined implicitly by other DNNs or observations like the time to memorisation. Our results connect with simplicity bias by identifying interpretable properties of the data that make samples appear "simple" to a shallow network. Interestingly, our findings reveal that different phases of learning experience simplicity in different ways, leading to forgetting of previously learned features.

**Spurious correlations.** Simplicity bias can also lead to shortcomings [39] by excessively relying of spurious features in the data, possibly hurting generalisation, especially in out-of-distribution contexts [14]. Theoretical works [29, 37, 18] have identified statistical properties that cause a classifier to favour spurious features over potentially more complex but more predictive features.

Various methods have been proposed to address this problem using explicit partitioning of the data [2, 36]; some approaches implicitly infer subgroups with various degrees of correlation as spurious features. Notably, [26, 42] rely on early stages of learning to detect bias and adjust sample importance accordingly. Our study provides a unifying view of learning in fairness and spurious correlation problems, highlighting the presence of ephemeral biases characterised by multiple timescales during training. This adds complexity to the understanding of learning dynamics and points out potential confounding effects in existing mitigation methods.

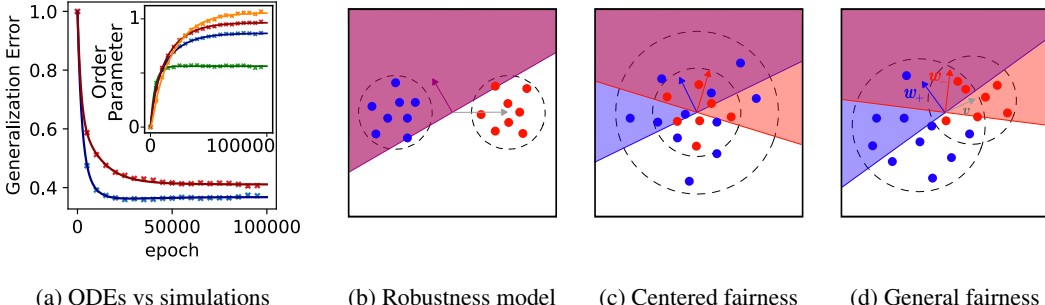

(a) ODEs vs simulations  (b) Robustness model  (c) Centered fairness  (d) General fairness

Figure 1: **Teacher-Mixture in fairness and robustness.** *Panel (a)* shows the generalisation errors— for the subpopulations $+$ (blue) and $-$ (red)—obtained through simulation (crosses) and predicted by the theory (solid lines) for a network with linear activation. The inset shows the same comparison for the *order parameters*: $R_+$ (blue), $R_-$ (red), $M$ (green), and $Q$ (orange). *Panels (b-d)* exemplify the different scenarios achievable in the TM model investigated in Sec. 4. *Panel (b)* represent a model for robustness where a spurious feature—given by the shift vector—can mislead the classifier, see Sec. 4.1. *Panels (c,d)* are instead discussed in Sec. 4.2 and represent two models of fairness. First, *Panel (b)* has no shift, $v = 0$, allowing us to remove the confounding effects. Finally, *Panel (d)* shows the general fairness problem.

## 2 Problem setup

**Data distribution.** We consider a standard supervised learning setup where the training data consists of pairs of a feature vector $\boldsymbol{x} \in \mathbb{R}^d$ and a binary label $y = \pm 1$. To model subgroups within the data [35], we assume that the feature vectors are structured as clusters $c_1, \ldots, c_m$, respectively centered on some fixed attribute vectors $\boldsymbol{v}_1, \cdots, \boldsymbol{v}_m \in \mathbb{R}^d$. Specifically, $\boldsymbol{x}$ is sampled from a mixture of $m$ isotropic Gaussians:

$$\boldsymbol{x} \sim \sum_{j=1}^{m} \rho_j \, \mathcal{N}(\boldsymbol{v}_j/\sqrt{d}, \Delta_j \mathbb{I}_{d \times d}), \tag{1}$$

with mixing probabilities $\rho_1, \cdots, \rho_m$ and scalar variances $\Delta_1, \cdots, \Delta_m$. Assuming the entries of $\boldsymbol{v}_j$ are of order 1 as $d$ gets large, the scaling factor $1/\sqrt{d}$ ensures that the Euclidean norm of the renormalised vector is of order 1. This prevents the problem from becoming either trivial or overly challenging in the high-dimensional limit [25, 24]. We adopt a teacher-mixture (TM) scenario [38] where each cluster has its own teacher rule:

$$\boldsymbol{x} \in c_j \quad \implies \quad y = \text{sign}(\overline{\boldsymbol{w}}_j^\top \boldsymbol{x}/\sqrt{d}). \tag{2}$$

This rule is characterised by the teacher vectors $\overline{\boldsymbol{w}}_j \in \mathbb{R}^d$, ensuring linear separability within each cluster. Fig. 1b-d illustrate the data distribution for two clusters with opposite mean vectors $\pm \boldsymbol{v}$, which will be the primary case study for our analysis.

**Model.** In this study we analyse a linear model applied to the above data distribution. We aim to learn a vector parameter $\boldsymbol{w}$, referred to as the 'student', such that predictions are given by

$$\hat{y}(\boldsymbol{x}) = \boldsymbol{w}^\top \boldsymbol{x}/\sqrt{d}. \tag{3}$$

The training process involves applying online SGD on the squared loss $\hat{\epsilon} = (y - \hat{y})^2$. At the $k$-th iteration, a feature vector $\boldsymbol{x}^k$ is sampled from (1), the ground truth label $y^k$ and current model

prediction $\hat{y}^k$ are respectively given by (2) and (3), and the parameter is updated as:

$$\Delta \boldsymbol{w}^k := \boldsymbol{w}^{k+1} - \boldsymbol{w}^k = -\frac{\eta}{2}\nabla\hat{\epsilon}^k(\boldsymbol{w}^k) = \frac{\eta}{\sqrt{d}}(y^k - \hat{y}^k)\boldsymbol{x}^k \quad (4)$$

where $\eta/2 > 0$ denotes the learning rate. It is important to note that in this online setting the number of time steps is equivalent to the number of training examples. In our analysis, the model is evaluated by its generalisation error, or population loss, $\epsilon := \mathbb{E}[\hat{\epsilon}]$.

## 3 SGD analysis

We study the evolution of the generalisation error during training with SGD with constant learning rate in the high dimensional setting (i.e. large $d$). Following a classical approach [34, 8], we streamline the problem by focusing on a small set of summary statistics, referred to as 'order parameters', which fully characterises the dynamics. As the dimension increases, it can be shown by concentration arguments that the evolution of these order parameters converges to the deterministic solution of a system of ODEs [15, 6, 3]. Notably, in our setting, we achieve an analytical solution of this ODE system. We sketch our main results below, referring to the Appendix for derivations and proofs.

### 3.1 Order parameters

In the setup described in Section 2, consider the following $2m + 1$ variables:

$$R_j = \frac{1}{d}\boldsymbol{w}^\top\overline{\boldsymbol{w}}_j, \quad M_j = \frac{1}{d}\boldsymbol{w}^\top\boldsymbol{v}_j, \quad Q = \frac{1}{d}\|\boldsymbol{w}\|^2, \quad (5)$$

for $1 \leq j \leq m$. These variables correspond to key statistics of the student, namely its alignment to the cluster teachers, its alignment to the cluster centers, and its magnitude, respectively.

**Lemma 3.1.** *The generalisation error can be written as an average $\epsilon = \sum_{j=1}^m \rho_j\epsilon_j$ over the clusters, where $\epsilon_j$ is a degree 2 polynomial in $R_j, M_j$ and $Q$ taking the form*

$$\epsilon_j = 1 - 2\alpha_j M_j + M_j^2 - \beta_j R_j + Q\Delta_j \quad (6)$$

*where $\alpha_j, \beta_j$ are constants independent of the parameter $\boldsymbol{w}$.*

We present the derivation of this result and the explicit form of the constants $\alpha_j, \beta_j$ in Appendix B.1.

Our problem thus reduces to characterising the evolution of order parameters (5). Using the gradient update of the parameter in Eq. 4 and the notation $\delta^k := y^k - \hat{y}^k$, we can write update equations for the order parameters as follows:

$$\Delta M_j^k = \frac{\eta}{d}\delta^k \frac{\boldsymbol{v}_j^\top \boldsymbol{x}^k}{\sqrt{d}}, \quad \Delta R_j^k = \frac{\eta}{d}\delta^k\frac{\overline{\boldsymbol{w}}_j^\top \boldsymbol{x}^k}{\sqrt{d}}, \quad \Delta Q^k = \frac{2\eta}{d}\delta^k\frac{\boldsymbol{w}_j^\top \boldsymbol{x}^k}{\sqrt{d}} + \frac{\eta^2}{d^2}(\delta^k)^2\|\boldsymbol{x}^k\|^2. \quad (7)$$

### 3.2 High dimensional dynamics

We build upon classic results [34, 8], recently put on rigorous grounds [15, 6, 3], leveraging the *self-averaging* property of the order parameters in the high dimensional limit $d \to \infty$. As a result, as the dimension gets large, the discrete, stochastic evolution (7) of the order parameters can be effectively described in terms of the deterministic solution of the average continuous-time dynamics.

Let $\mathcal{S} := (S_i)_{1 \leq i \leq 2m+1}$ denote the collection of order parameters. The following lemma shows that the average of the updates (7) over the sample $\boldsymbol{x}^k$ can be expressed solely in terms of $\mathcal{S}^k$.

**Lemma 3.2.** $\mathbb{E}[\Delta S_i^k] = \frac{1}{d}f_i(\mathcal{S}^k)$ *for some functions $(f_i(\mathcal{S}))_{1 \leq i \leq 2m+1}$ in O(1) as $d \to \infty$.*

The theorem below states that as $d$ gets large, the stochastic evolution $\mathcal{S}^k$ of the order parameter gets uniformly close, with high probability, to the average continuous-time dynamics described by the ODE system:

$$\frac{d\bar{\mathcal{S}}_i(t)}{dt} = f_i(\bar{\mathcal{S}}(t)), \quad 1 \leq i \leq 2m+1, \quad (8)$$

where the continuous *time* is given by the example number divided by the input dimension, $t = k/d$. Formally,

**Theorem 3.3.** *Fix a time horizon $T > 0$. For $1 \leq i \leq 2m + 1$,*

$$\max_{0 \leq k \leq dT} |\mathcal{S}_i^k - \bar{\mathcal{S}}_i(k/d)| \xrightarrow{P} 0 \quad \text{as } d \to \infty. \tag{9}$$

where $\xrightarrow{P}$ denotes convergence in probability. A proof is provided in Appendix B. We provide the explicit expression of the functions $f_i$ in the ODEs (8) in Appendix C, focusing on $m = 2$ clusters for clarity.

## 3.3 Solving the ODEs

Here we present the explicit solution of the ODEs (8) in the case of two clusters ($m = 2$) with opposite mean vectors $\pm v$, as in [38]. Henceforth, we refer to $v$ as the shift vector and to the two clusters as the 'positive' and 'negative' sub-populations, with mixing probabilities $\rho$ and $(1 - \rho)$, variances $\Delta_\pm$ and teacher vectors $\overline{w}_\pm$, respectively. The order parameters introduced in Eq. 5 are specifically denoted as $M = w^\top v/d$, $R_+ = w^\top \overline{w}_+/d$, and $R_- = w^\top \overline{w}_-/d$ in this setting.

**Theorem 3.4.** *In the above setting, solutions to the order parameter evolution take the form*

$$M(t) = M_0 e^{-\eta(v+\Delta^{mix})t} + M^\infty(1 - e^{-\eta(v+\Delta^{mix})t}), \tag{10}$$

$$R_\pm(t) = R_\pm^0 e^{-\eta\Delta^{mix}t} + R_\pm^\infty(1 - e^{-\eta\Delta^{mix}t}) + k_1^\pm(e^{-\eta\Delta^{mix}t} - e^{-\eta(v+\Delta^{mix})t}), \tag{11}$$

$$\begin{aligned} Q(t) &= Q_0 e^{-\eta(2\Delta^{mix}-\eta\Delta^{2mix})t} + Q^\infty(1 - e^{-\eta(2\Delta^{mix}-\eta\Delta^{2mix})t}) \\ &+ k_2(e^{-t(2\Delta^{mix}-\eta\Delta^{2mix})\eta} - e^{-t\Delta^{mix}\eta}) + k_3(e^{-t(2\Delta^{mix}-\eta\Delta^{2mix})\eta} - e^{-t(v+\Delta^{mix})\eta}) \\ &+ k_4(e^{-t(2\Delta^{mix}-\eta\Delta^{2mix})\eta} - e^{-t(2v+2\Delta^{mix})\eta}), \end{aligned} \tag{12}$$

*with $\Delta^{mix} = \rho\Delta_+ + (1 - \rho)\Delta_-$, $\Delta^{2mix} = \rho\Delta_+^2 + (1 - \rho)\Delta_-^2$ and $v = ||v||^2/d$.*

The remaining constants are less significant and are reported in Appendix D.1 and discussed further in Appendix E. This solution allows us to describe important observables such as the generalisation error (via Lemma 3.1) at any timestep. Fig. 1a plots the theoretical closed-form solutions along with values obtained through simulation when we set $d = 1000$. Note the remarkable agreement between the analytical ODE solution and simulations of the online SGD dynamics in this high dimensional data limit.

## 4 Insights

In this section, we delve deeper into the solution derived in Theorem 3.4. By examining the exponents in Eqs. 10-12, we can identify the relevant training timescales. Notably, $M$ follows a straightforward behaviour dominated by a single timescale, whereas $R_\pm$ and $Q$ exhibit multiple timescales, leading to significant implications for the emergence and evolution of bias during training.

In the following sections, we analyse increasingly complex scenarios to understand how bias develops and evolves. Parameters specifying these different scenarios are the shift norm $v = ||v||^2/d$ and relative representation $\rho$, the subpopulation variances $\Delta_\pm$, and the teacher overlap $T_\pm = \overline{w}_+^\top \overline{w}_-/d$. For simplicity we fix the teacher norm $||w_\pm||_2 = \sqrt{d}$, so that $T_\pm$ is the cosine similarity between the two teachers.

### 4.1 Spurious correlations

The emergence of spurious correlations during training exemplifies a type of bias where a classifier favours a spurious feature over a core one. To isolate the impact of spurious correlation in our model while

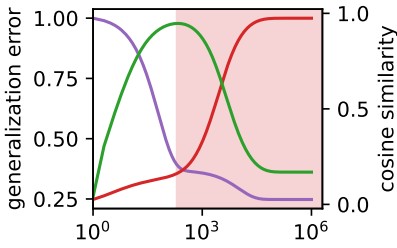

Figure 2: **Spurious correlations transient alignment.** Time-evolution of loss (purple), student-teacher (red) and student-shift (green) cosine similarities. The initial phase (green background) of learning aligns classifier and shift vector before aligning with the teacher (red background), Sec. 4.1. Parameters: $v = 16, \rho = 0.5, \Delta_- = \Delta_+ = 0.1, T_\pm = 1, \eta = 0.5$. For these parameters, spurious features allow the correct classification of 90% of the samples.

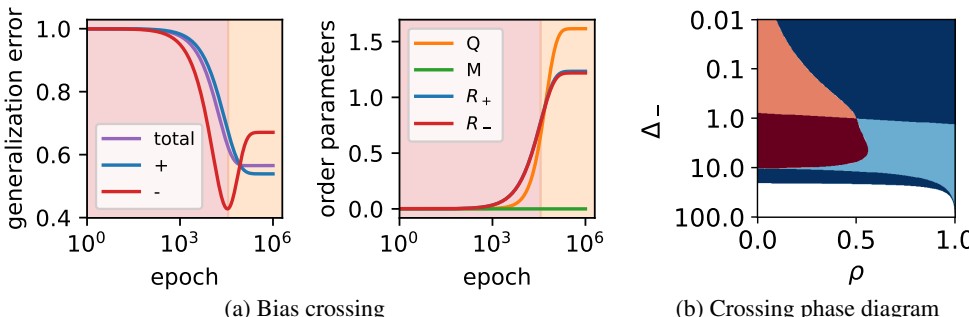

Figure 3: **The crossing phenomenon.** *Panel (a) (left side)* shows the loss curves of sub-population $-$ (in red) and sub-population $+$ in blue along with the overall loss (in purple). We observe a crossing cause by a higher variance but lower representation in sub-population $-$. The background colours represent the different phases of bias that are characterised by the evolution of the order parameters shown in *Panel (a) (right side)*. *Panel (b)* shows the presence of the crossing phenomenon in a large portion of the parameter space using a phase diagram. Blue indicates an asymptotic preference for sub-population $+$ and red the opposite. Dark colours indicates regions where bias is consistent across training, while regions in light colours undergo a crossing phenomenon. White indicates that learning rate was too high and training diverged. Parameters: $v = 0, \Delta_+ = 1, T_\pm = 0.9, \eta = 0.1$.

avoiding confounding effects, we consider perfectly overlapping teachers ($\overline{w}_+ = \overline{w}_-$) and sub-populations with equal variance and representation ($\rho = 0.5, \Delta_+ = \Delta_-$). With non-perfectly overlapping clusters $v \neq 0$, we introduce a spurious correlation by adding a small cosine similarity between the shift vector and the teacher, creating a label imbalance—an imbalance between the proportion of positive and negative labels—within each sub-population. The setting is illustrated in Fig. 1b.

From Eqs. 10-12, two relevant timescales for the problem are observed:

$$\tau_M = \frac{1}{\eta(v + \Delta^{mix})}, \qquad (13) \qquad\qquad \tau_R = 1/\eta\Delta^{mix}. \qquad (14)$$

The shortest timescale, $\tau_M$, associated with $M$, indicates that the student first aligns with the spurious feature. By aligning with the shift vector, the student can predict most examples correctly, but not all. The effect of spurious correlations is transient; at $t \sim \tau_R$, the student starts disaligning from the spurious feature and aligns with the teacher vector, eventually achieving nearly perfect alignment. This is illustrated in Fig. 2, where the student initially picks up on the spurious correlation (green) and achieves almost perfect alignment with the shift vector during intermediate times before aligning nearly perfectly with the teacher (red).

## 4.2 Fairness

In this section, we identify the properties of sub-populations that determine the bias during learning and show how bias evolves in three phases. To quantify bias, we use the *overall accuracy equality* metric [7], which measures the discrepancy in accuracy across groups. Intuitively, we aim for equal loss on both groups, considering any deviation from this condition as bias.

### 4.2.1 Zero shift

We first consider a simplified case where we assume that both clusters are centered at the origin $v = 0$ as shown in Fig. 1c. We will later reintroduce the shift and analyse the transient dynamics it introduces as per the discussion in section 4.1. The zero shift case represents an extreme situation where not only is the classification not solvable if $\overline{w}_+ \neq \overline{w}_-$, but it is also difficult to identify which cluster generated a given data point since the shift provides no information. This setting is particularly suited to analysing the effects of 'group level' features, such as group variance and relative representation, on the preference of the classifier.

In this simplified setting, $M(t)$ is always zero and the constants $k_1^{\pm}, k_3, k_4$ presented in equations 11 and 12 are zero. Thus, the dynamics only involve two relevant timescales given by $\tau_R$ in Eq. 14 and

$$\tau_Q = 1/(\eta(2\Delta^{mix} - \eta\Delta^{2mix})). \tag{15}$$

Fig. 3a illustrates the changing preference of the classifier. Specifically, we observe that the variance of the sub-population is particularly relevant initially and the sub-population with higher variance (red) is *learnt* faster, i.e. its generalisation error drops faster. However, asymptotically we observe that the relative representation becomes more important wherein the student aligns itself with the teacher that has a higher product of representation and standard deviation (blue), i.e.

$$\rho\sqrt{\Delta_+} \gtrless (1-\rho)\sqrt{\Delta_-} \iff R_+^{\infty} \gtrless R_-^{\infty}. \tag{16}$$

Thus, the network can advantage the cluster with higher variance initially but asymptotically advantage the other cluster if its representation is high enough. This leads to the interesting behaviour shown in Fig. 3 wherein we observe a 'crossing' of the losses on the two sub-populations. A more detailed analysis of the 'crossing' is presented in Appendix E.2.

*Initial dynamics.* Starting from small initialisation, the initial rate of change of the generalisation error for sub-population $+$ is

$$\frac{d\epsilon_{g+}}{dt}\bigg|_{t=0} = -\eta^2\Delta^{mix}\Delta_+ \left( \sqrt{\frac{2}{\pi\Delta_+}}\frac{R_+^{\infty}}{\eta} - 1 \right) \tag{17}$$

and analogously for $-$. The learning rate $\eta$ must be chosen to be small enough such that the generalisation errors decrease and hence the first term in the brackets must dominate over the 1. Since $R_+^{\infty}/R_-^{\infty} \in [T_{\pm}; 1/T_{\pm}]$ (for $T_{\pm} > 0$), the ratio between generalisation error rates is bounded by

$$T_{\pm}\sqrt{\frac{\Delta_+}{\Delta_-}} \leq \frac{d\epsilon_{g+}/dt|_{t=0}}{d\epsilon_{g-}/dt|_{t=0}} \leq \frac{1}{T_{\pm}}\sqrt{\frac{\Delta_+}{\Delta_-}}. \tag{18}$$

When the teachers are only slightly misaligned—$T_{\pm} \lesssim 1$—the bound is tight and we can see that it is the ratio of the square roots of the variances that determines which cluster is learnt faster initially. As precisely detailed below, the initial bias can substantially differ from the asymptotic bias of the classifier. Indeed, Fig. 3b shows in a phase diagram the existence of 'bias crossing' across a wide range of variances and representations. The transition between the phases that represent a initial preference for the positive sub-population (light red and dark blue) and the phases that represent an initial preference for negative sub-population (dark red and light blue) is approximately given by the line $\Delta_- = \Delta_+ = 1$, independent of the representation as predicted by Eq. 18. The portion of the dark blue phase just above the white divergent phase marks a 'quasi-divergent' region wherein the generalisation error on the negative sub-population rises even at $t = 0$ because the learning rate is too large for such high variances. It hence marks a region of impractical behaviour that is only observed with poorly optimised learning rates.

*Asymptotic preference.* In the limit of small learning rates $\eta \to 0$, the student will asymptotically exhibit lower loss on whichever sub-population's teacher it has better alignment with. Thus, Eq. 16 provides a simple characterisation of asymptotic preference from representations and standard deviations in the small learning rate limit. However, the situation is more complex in the case of finite learning rate, which may disrupt learning in one or both clusters. Without indulging further into the discussion of asymptotic performance, which is not the main goal of the paper, we refer to Appendix E.3 for more in depth and analysis and additional phase diagrams.

### 4.2.2 General case

We now consider the general case shown in Fig. 1d, where the shift is non zero and all three timescales identified so far play a role.

As observed in Sec. 4.1, when the shift norm $v$ is large, the effect of spurious correlations becomes significant and the timescale associated with the spurious correlations is the fastest. In general, when $v \neq 0$ we observe an additional phase due to the effect of spurious correlation. In this new first phase, the student advantages the cluster with higher representation and lower variance since the salient information received from this cluster is more coherent and easier to access.

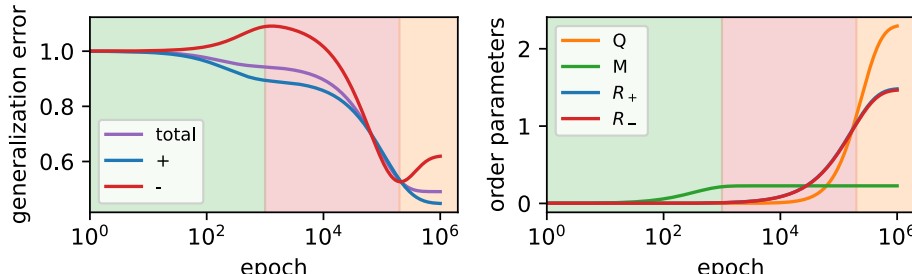

Figure 4: **Double crossing phenomenon.** *(Left panel)* shows the loss for the two sub-populations (blue and red lines) and the global one (in purple). *(Right panel)* shows the value of the order parameters across time. The behaviour of the order parameters across time provides a precise characterisation and understanding of the different phases. Parameters: $v = 100, \rho = 0.75, \Delta_+ = 0.1, \Delta_- = 0.5, \eta = 0.03, T_\pm = 0.9, \alpha_+ = 0.343, \alpha_- = 0.12$.

More precisely, in high dimensions the shift and the teachers are likely to exhibit a small cosine similarity leading to a class imbalance in the clusters and creating spurious correlation. The amount of label imbalance within a cluster is characterised by the value of $\alpha$, as detailed in Appendix A. For smaller variances, $\alpha$ takes more extreme values leading to stronger spurious correlation of that cluster with the shift. If a cluster has more positive examples, we would observe a reduction in loss for that cluster if the student aligns with the mean of that cluster (and opposite to the mean if the cluster has mostly negative examples). When both clusters have different majority classes, the direction of spurious correlation for the two are same. However, when the majority classes are the same, we have competing directions for spurious correlation. The expression for $M_\infty$ in Appendix D.1 Eq. D.42 shows that in this case the relative representation comes into play and the mean of the cluster with greater representation and class imbalance will be chosen by the teacher to align with. Fig. 4 shows such a scenario with three phase bias evolution:

- The green phase is driven by spurious correlation where the positive cluster is advantaged since it has greater representation and class imbalance.

- Then, the red phase is driven by greater variance where the negative cluster is learnt faster as discussed through Eq. 18.

- Finally, we observe the orange phase wherein the student starts aligning with the positive cluster as per the asymptotic rule in Eq. 16.

In summary, the student first advantages the sub-population with higher representation and lower variance. Next, it advantages the sub-population with higher variance. Asymptotically, it advantages the sub-population with higher representation and variance. Our analysis thus shows that bias is a dynamical quantity that can vary non-monotonically during training and cannot be characterised by simply the initial and asymptotic values.

## 5 Ablations using numerical simulations

### 5.1 Rotated MNIST

We train a 2-layer neural network with 200 hidden units, ReLU activation, and sigmoidal readout activation, using online SGD on MSE loss in a MNIST classification task. Data are centered and the variance is set to 1 following standard pre-processing practices. We consider a variation of the MNIST dataset that mimics the TM model, allowing us to verify our theory when network and the data structure are mismatched. Digits 0 to 4 and 5 to 9 are grouped to form the two subpopulations. With probability $p_+$ and $p_-$, digits of both subpopulations are rotated with a subpopulation-specific angle—i.e. Fig. 5a uses angles of rotation $\theta_- = 45^o$ and $\theta_- = -90^o$. The goal of the classifier is to detect rotations.

The experimental framework gives a correspondence between parameters of the generative model and properties of a real dataset. We can control relative representation by subsampling, teacher similarity by playing with angle difference, label imbalance by changing the probability of rotation, and saliency by increasing and decreasing the norm of the subpopulation using multiplicative factors $\Delta_\pm$. The only parameter that we cannot control is the shift $v$ which is a property of the data.

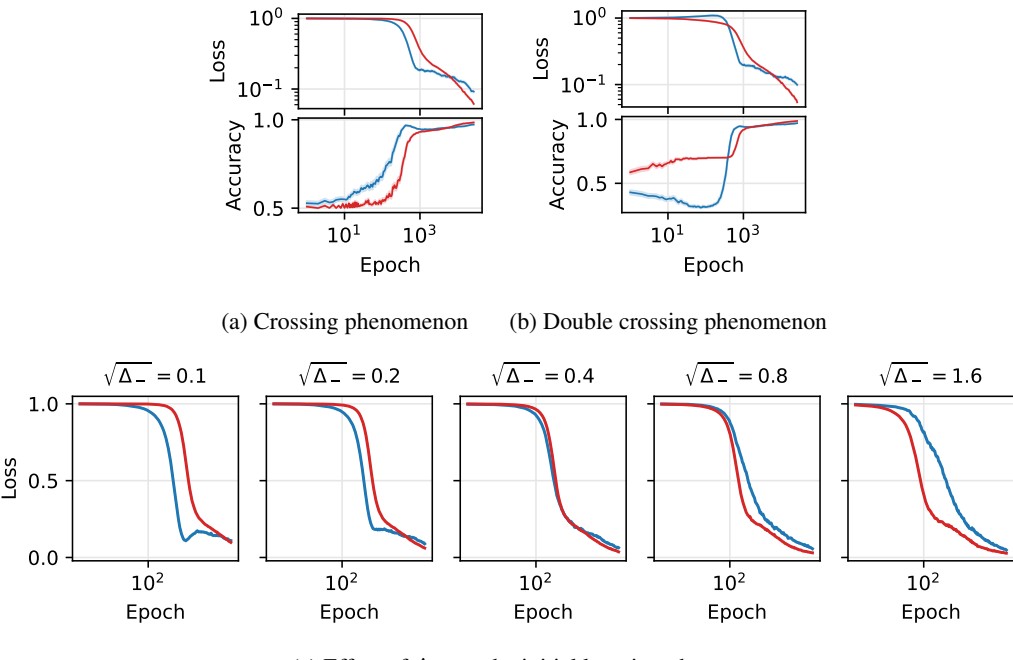

(a) Crossing phenomenon     (b) Double crossing phenomenon

(c) Effect of $\Delta_-$ on the initial learning phase

Figure 5: **Numerical simulations on MNIST.** The figure shows the average (solid lines) and standard deviation (shaded area) of 100 simulations run in this framework. In particular the upper plots show the test loss and lower plots the test accuracy for subpopulation $+$ (blue) and $-$ (red). *Panel (a)* an example of crossing phenomenon obtained by imposing $\sqrt{\Delta_+} = 1$, $\sqrt{\Delta_-} = 0.2$, and $\rho = 0.1$. *Panel (b)* shows the double crossing, obtained by introducing an additional timescale to the previous case by tuning label imbalance. *Panel (c)* explore the effect of changing $\Delta_-$ while keeping a constant $\Delta_+ = 1$.

Therefore, in order to reproduce the zero-shift case of Sec. 4.2, we remove the label imbalance by setting the probability of rotation $p_+ = p_- = 0.5$. By properly calibrating the saliency $\Delta$ and the relative representation $\rho$, it is possible to bias the classifier towards one subpopulation at the beginning of training and the other in the end. This is shown in Fig. 5a where $\rho = 0.1$ and $\Delta_+ > \Delta_-$. The saliency difference favours subpopulation $+$ initially while setting $\rho$ small enough advantages subpopulation $-$ later in training. This is precisely what we observe in the plot.

In Fig. 5b, we extend the MNIST experiment by varying the average image brightness of subpopulation $-$, which reflects the group variance in our theory. The results show that greater brightness leads to faster learning in the second phase and an increasing asymptotic preference, consistent with our predictions.

Finally, we consider the general fairness case. By creating label imbalance, i.e. setting $p_+ = 0.3$ and $p_- = 0.7$, we observe an additional phase of bias evolution, wherein the classifier prefers dense regions with consistent labels. This advantages subpopulation $-$ and indeed it is what we see in Fig. 5c. The result of the simulations matches the theory displaying a double crossing phenomenon.

## 5.2  CIFAR10

We consider the same architecture and pre-processing described for MNIST on a CIFAR10 classification task. We select 8 classes and assign 4 of them to the positive group and 4 to the negative group. Inside each group, 2 classes are labelled as negative and 2 as positive. This simulation framework is similar to the one considered by [5] where the authors used sub-populations with only 2 classes each.

The average brightness of the samples in each cluster plays the same role as the parameter $\Delta$ in the synthetic model. Our theory predicts that the classifier will advantage the group with highest average brightness, see Eq. 16. In order to achieve the same generalisation error on both subpopulations, the less bright group needs more samples (larger $\rho$). This is shown in Fig. 6a, where the three panels correspond to different assignment of the classes: in the top panel classes are randomly assigned to

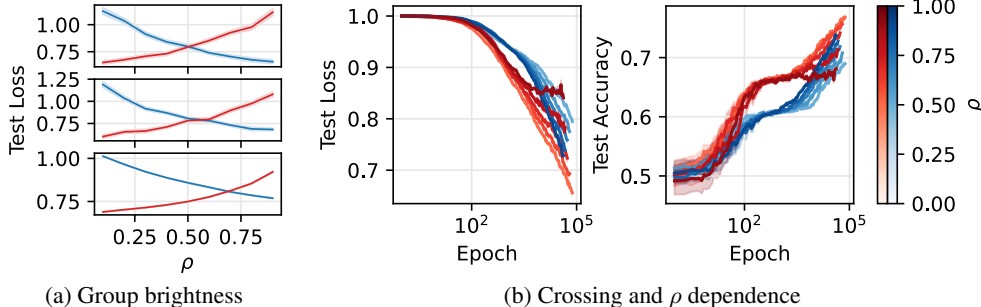

(a) Group brightness             (b) Crossing and $\rho$ dependence

Figure 6: **Numerical simulations on CIFAR10.** The figure shows experiments of a 2L neural network on CIFAR10 where classes were grouped together to form the subpopulations. The plots show the average performance—measure by loss or accuracy—achieved over 100 simulations (for *Panel (a)*) and 10 simulations (for *Panel (b)*, respectively) using the shaded area to quantify the standard deviation. *Panel (a)* shows the result at the end of training changing relative representation $\rho$, while *Panel (b)* shows the training trajectories as $\rho$ changes as indicated in the colour-bar, see text for more details.

the two groups; in the middle panel classes are randomly partitioned in two groups and the brighter one is assigned to group $-$; finally the last panel assigns the brightest classes to group $-$ and least bright to group $+$. As predicted, we need increasingly high relative representation $\rho$ to achieve a balance in losses at the end of training.

When labels are balanced, our theory predicts that the classifier is initially attracted by the larger $\Delta$ and eventually—if the relative representation of the group with smaller $\Delta$ is large enough—it switches and favours the other group. This effect is indeed verified in the CIFAR10 experiments. Starting from the partitioning in Fig. 6a (bottom) with $\rho = 0.8$, the dynamics is initially attracted by group – before advantaging the other group, giving rise to a crossing that can be observed–among other things–in Fig. 6b.

Fig.6b highlights another prediction of our theory concerning the timescale when $\rho$ becomes relevant. Our theory predicts that $\rho$ should become relevant only in the final stages of the dynamics and indeed the curves are almost perfectly overlapping until for most of training. As already noticed, $\rho$ sufficiently large gives rise to a crossing behaviour as indicated demonstrated in the synthetic model.

## 5.3 Additional numerical experiments

In Appendix F, we provide additional experiments within the TM model and the CelebA dataset, exploring different architectures and losses. Even under these new conditions, we observe that bias presents different timescales and shows crossing behaviours.

## 6 Conclusion

This paper examined the dynamics of bias in a high dimensional synthetic framework, showing that it can be explicitly characterised to reveal transient behaviour. Our findings reveal that classifiers exhibit biases toward different data features during training, possibly alternating sub-population preference. Although our analysis is based on certain assumptions, numerical experiments that violate these assumptions still display the behaviour predicated by our theory.

While this paper centered on bias propagation in a controlled synthetic setting, the study of bias in ML systems is a significant issue with profound societal implications. We believe this line of research will have practical impacts in the medium term, aiding the design of mitigation strategies that account for transient dynamics. Future research will further explore this connection, proposing theory-based dynamical protocols for bias mitigation.

## Acknowledgements

We thank Andrew Saxe and Simon Lacoste-Julien for numerous discussions and insightful comments.Anchit Jain is grateful to the Gatsby Computational Neuroscience Unit for the research internship opportunity. This work was supported by the Sainsbury Wellcome Centre Core Grant from Wellcome (219627/Z/19/Z) and the Gatsby Charitable Foundation (GAT3755).

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

# Appendix

**Contents**

# A  Problem setup and notation

We begin by refreshing the problem description and notation introduced in the main body for the two cluster case (Sec. 3.3) as well as defining some new notation to make the presentations of the results more compact.

1. $(\boldsymbol{x}, y)$ denotes a training example with $\boldsymbol{x} \in \mathbb{R}^d$ and $y \in \{-1, 1\}$.

2. $\boldsymbol{x}$ is drawn from a mixture of two Gaussians with means $\boldsymbol{v}/\sqrt{d}$ and $-\boldsymbol{v}/\sqrt{d}$ respectively, covariances $\Delta_+ I_{d \times d}$ and $\Delta_- I_{d \times d}$ respectively. These two Gaussians are henceforth referred to as the positive and negative Gaussians respectively.

3. $\rho$ represents the probability of the data being drawn from the positive Gaussian.

4. $\langle \rangle$ denotes an average over $x$, $\langle \rangle_\oplus$ denotes an average over the positive Gaussian and $\langle \rangle_\ominus$ denotes an average over the negative Gaussian.

5. $\overline{\boldsymbol{w}}_+$ and $\overline{\boldsymbol{w}}_-$ denote the teachers for the positive Gaussian and negative Gaussian respectively. $\boldsymbol{w}$ is the learnt classifier ("the student").

6. The true labels, $y$, are then given by:
   - $y = \text{sign}(\overline{\boldsymbol{w}}_+ \cdot \boldsymbol{x}/\sqrt{d})$ for the positive cluster;
   - $y = \text{sign}(\overline{\boldsymbol{w}}_- \cdot \boldsymbol{x}/\sqrt{d})$ for the negative cluster.

7. Our predictions are $\hat{y} = \boldsymbol{w} \cdot \boldsymbol{x}/\sqrt{d}$.

8. The student is trained to minimise L2 loss = $(y - \hat{y})^2$.

9. The student learns using online stochastic gradient descent.

10. $\eta/2$ is the learning rate.

11. $\epsilon$ denotes the generalisation error.

12. $a \cdot b$ denotes the dot product between vectors $a$ and $b$.

13. We now define the following Order Parameters (where only the first 4 change with training):
    - $Q = \boldsymbol{w} \cdot \boldsymbol{w}/d$;
    - $R_+ = \boldsymbol{w} \cdot \overline{\boldsymbol{w}}_+/d$;
    - $R_- = \boldsymbol{w} \cdot \overline{\boldsymbol{w}}_-/d$;
    - $M = \boldsymbol{w} \cdot \boldsymbol{v}/d$;
    - $T_\pm = \overline{\boldsymbol{w}}_+ \cdot \overline{\boldsymbol{w}}_-/d$;
    - $M_+^* = \overline{\boldsymbol{w}}_+ \cdot \boldsymbol{v}/d$;
    - $M_-^* = \overline{\boldsymbol{w}}_- \cdot \boldsymbol{v}/d$;
    - $v = \boldsymbol{v} \cdot \boldsymbol{v}/d$.

14. For algebraic simplicity, we assume $\|\overline{\boldsymbol{w}}_+\|_2 = \|\overline{\boldsymbol{w}}_-\|_2 = \sqrt{d}$ (and thus, $\overline{\boldsymbol{w}}_+ \cdot \overline{\boldsymbol{w}}_+/d = 1$ and $\overline{\boldsymbol{w}}_- \cdot \overline{\boldsymbol{w}}_-/d = 1$). This has the consequence that $T_\pm$ exactly equals the cosine similarity between the two teachers.

15. We also define $\Delta^{mix} = \rho\Delta_+ + (1-\rho)\Delta_-$ and $\Delta^{2mix} = \rho\Delta_+^2 + (1-\rho)\Delta_-^2$.

16. For notational convenience we define:

$$\alpha_+ = \langle y \rangle_\oplus = 1 - 2\Phi\left(\frac{-M_+^*}{\sqrt{\Delta_+}}\right), \tag{A.19}$$

$$\alpha_- = \langle y \rangle_\ominus = 1 - 2\Phi\left(\frac{-(-M_-^*)}{\sqrt{\Delta_-}}\right). \tag{A.20}$$

Note, $\alpha_+$ also has an intuitive meaning. It represents the difference between the probability that an example drawn from the positive cluster has positive true label and the probability that an example drawn from the positive cluster has negative true label. It is hence 0 when the positive cluster has equal positive and negative examples, positive when the cluster has more positive examples than negative, negative when the cluster has more negative examples than positive. Similarly, $\alpha_-$ represents the difference in these probabilities for the negative cluster.

17. Finally, we also define

$$\beta_+ = \sqrt{\frac{2\Delta_+}{\pi}} \exp\left(\frac{-M_+^{*2}}{2\Delta_+}\right), \tag{A.21}$$

$$\beta_- = \sqrt{\frac{2\Delta_-}{\pi}} \exp\left(\frac{-M_-^{*2}}{2\Delta_-}\right). \tag{A.22}$$

18. Lastly, we use $t$ to denote continuous time given by (epoch number/$d$).

# B  Proof of the main theorems

## B.1  Proof of Lemma 3.1.

Denote with $\langle\cdot\rangle_j$ the expectation over samples from cluster $j$. The generalisation error reads $\epsilon = \sum_{j=1}^m \rho_j \epsilon_j$ with

$$\epsilon_j := \langle(y-\hat{y})^2\rangle_j = \left\langle\left(y - \frac{\boldsymbol{w}\cdot\boldsymbol{x}}{\sqrt{d}}\right)^2\right\rangle_j = \langle y^2\rangle_j + \left\langle\left(\frac{\boldsymbol{w}\cdot\boldsymbol{x}}{\sqrt{d}}\right)^2\right\rangle_j - 2\left\langle y\frac{\boldsymbol{w}\cdot\boldsymbol{x}}{\sqrt{d}}\right\rangle_j$$

$$= 1 + (Q\Delta_j + M_j^2) - 2(\alpha_j M_j + R_j\beta_j),$$

where the second term comes from: isolating the mean and the definition of $M_j$, and the isotropy of $x$. The third term comes from the useful identity *Integral 1* Eq. C.31, derived in Appendix C.1, and the constants are given by

$$\alpha_j = 1 - 2\Phi\left(\frac{-M_j^*}{\sqrt{\Delta_j}}\right), \quad \beta_j = \sqrt{\frac{2\Delta_j}{\pi}} \exp\left(\frac{-(M_j^*)^2}{2\Delta_j}\right). \tag{B.23}$$

where $M_j^* := \overline{\boldsymbol{w}}_j^\top \boldsymbol{v}_j/d$ and $\Phi(x) = \frac{1}{\sqrt{2\pi}}\int_{-\infty}^x e^{-u^2/2}du$ is the cumulative distribution function of the standard normal.

The formula for the generalisation error specializes to the case of two clusters with opposite means as

$$\epsilon = 1 + M^2 - (2\rho\alpha_+ - 2(1-\rho)\alpha_-)M \\ - 2\rho\beta_+ R_+ - 2(1-\rho)\beta_- R_- + \Delta^{mix}Q, \tag{B.24}$$

Notably, $\alpha_\pm$ has an intuitive meaning wherein it represents the difference between the fraction of positive and negatives in a cluster, i.e., $\alpha_+ = \langle y\rangle_{c=+}$ and $\alpha_- = \langle y\rangle_{c=-}$.

## B.2  Proof of Lemma 3.2.

Explicit computations are carried out in Appendix C.2 below for the case of two clusters.

## B.3  Proof of Thm 3.3.

Using the notation of Section 3.2 and assuming Lemma 3.2, we examine the update equations (7) written as a stochastic iterative process

$$\mathcal{S}^{k+1} = \mathcal{S}^k + \mathbb{E}\frac{1}{d}f(\mathcal{S}^k) + \frac{1}{\sqrt{d}}\xi_d^k, \qquad \xi_d^k := \sqrt{d}(\Delta\mathcal{S}^k - \mathbb{E}[\Delta\mathcal{S}^k]) \tag{B.25}$$

where the expectation is over the new sample $\boldsymbol{x}^k$ and conditional on the past samples. The noise term $\xi_d^k$ has zero mean $\mathbb{E}[\xi_d^k] = 0$ and conditional covariance $\Sigma_d := \mathbb{E}[\xi_d^k\xi_d^{k\top}]$.

Define the continuous-time rescaled process $S_d(t)$) as the linear interpolation of $S^{\lfloor td\rfloor}$:

$$S_d(t) = S^{\lfloor td\rfloor} + (td - \lfloor td\rfloor)(S^{\lfloor td\rfloor+1} - S^{\lfloor td\rfloor}) \tag{B.26}$$

Here we leverage existing stochastic process convergence results (e.g., [6], Theorem 2.3]) showing that, if $\Sigma_d$ converges to the matrix valued function $\Sigma(\mathcal{S})$ as $d \to \infty$ in some appropriate sense, then

the sequence $S_d(t)$ converges weakly as $d \to \infty$ to the solution $\tilde{S}_t$ of the stochastic differential equation:

$$d\tilde{S}_t = f(\tilde{S}_t)dt + \sqrt{\Sigma(\tilde{S}_t)}dB_t \tag{B.27}$$

where $B_t$ is a standard Brownian motion in $\mathbb{R}^{2m+1}$. In our case, we can show that $\Sigma_d \in \mathcal{O}(d^{-1})$ as $d \to \infty$, so that $\Sigma = 0$ and Eq. B.27 reduces to the ODE in Eq. 8.

Let us sketch the scaling argument. Algebraic manipulations similar to those in Section C.2 show that

$$\Sigma_d = \nabla \mathcal{S}^{k\top} \mathbb{E}[\Phi^k \Phi^{k\top}] \nabla \mathcal{S}^k (1 + \mathcal{O}(d^{-1})), \qquad \Phi^k := \eta(\delta^k \boldsymbol{x}^k - \mathbb{E}[\delta^k \boldsymbol{x}^k]) \tag{B.28}$$

where $\nabla$ denotes the gradient with respect to the student vector $\boldsymbol{w}$. Recall that $S^k$ has $2m$ components that are linear in $\boldsymbol{w}$ (corresponding to the order parameters $R_j$ and $M_j$ in Eq. 5) and one that is quadratic (corresponding to $Q$). By making the gradients $\nabla \mathcal{S}^k$ explicit using Eq. 5), we see that at leading order, the matrix entries $\Sigma_d^{ij}, 1 \le i, j \le 2m + 1$ take the form

$$\Sigma_d^{ij} = \frac{1}{d}\mathbb{E}[\Phi_{\boldsymbol{a}_i}^k \Phi_{\boldsymbol{a}_j}^{k\top}], \qquad \Phi_{\boldsymbol{a}_i}^k = \eta(\delta^k \frac{\boldsymbol{a}_i^\top \boldsymbol{x}^k}{\sqrt{d}} - \mathbb{E}[\delta^k \frac{\boldsymbol{a}_i^\top \boldsymbol{x}^k}{\sqrt{d}}]) \tag{B.29}$$

where the vector $\boldsymbol{a}_i$ is either one of the teacher vectors $\overline{\boldsymbol{w}}_j$, one of the shift vector $\boldsymbol{v}_j$, or the student vector $\boldsymbol{w}$, depending on the entry $i = 1, \cdots, 2m + 1$. As can be shown explicitly as in Appendix C.1 below, $\Phi_{\boldsymbol{a}_i}^k$ depend on $\boldsymbol{x}^k$ only through auxiliary variables $\overline{\boldsymbol{w}}_j^\top \boldsymbol{x}/\sqrt{d}, \boldsymbol{v}_j^\top \boldsymbol{x}/\sqrt{d}, \boldsymbol{w}^{k\top}\boldsymbol{x}^k/\sqrt{d}$, which jointly follow a multivariate distribution whose parameters depend on the student vector $\boldsymbol{w}^k$ only through $\mathcal{S}^k$ and are in $O(1)$ as $d \to \infty$. As a result, $\Sigma_d^{ij} \in O(d^{-1})$.

Finally, the weak convergence of $S_d(t)_t$ to $\bar{S}_t$ implies convergence in probability for the supremum norm on the interval $[0, T]$ for any $T > 0$. Specifically, for each $1 \le i \le 2m + 1$,

$$\sup_{0 \le t \le T} |S_{di}(t) - \bar{S}_i(t)| \xrightarrow{P} 0, \tag{B.30}$$

where $\xrightarrow{P}$ denotes convergence in probability. This result directly leads to Eq. 9, thereby proving the theorem.

## C  Derivation of the ODEs

In this section we are going to explicitly derive the ODE describing the dynamics of the order parameters. Starting from the discrete updates of the order parameters, Eqs. 7, we are going to consider the thermodynamic limit, $d \to \infty$. As proven in Thm. 3.3, the updates concentrate to their typical value and the discrete evolution converges to differential equations. Therefore, the rest of the section is devoted to performing averages over the Gaussians in order to evaluate the typical values. Before proceeding with the evaluation of Eqs. 7, it is useful to introduce two identities.

### C.1  Useful Averages

Integral 1:

$$\langle a \cdot x \, \text{sign}(b \cdot x + c) \rangle = (a \cdot \mu)(1 - 2\Phi\left(\frac{-(b \cdot \mu + c)}{\sqrt{\Delta b \cdot b}}\right)) + a \cdot b \sqrt{\frac{2\Delta}{b \cdot b\pi}} \exp\left(\frac{-(b \cdot \mu + c)^2}{2\Delta b \cdot b}\right) \tag{C.31}$$

where $x$ is multivariate normal distribution with mean $\mu$ and covariance $\Delta I$, and the angular bracket notation indicates average with respect to $x$.

*Derivation.* Define the auxiliary random variables $z_1 = a \cdot x$ and $z_2 = b \cdot x + c$, that follow a multivariate normal distribution

$$\begin{bmatrix} z_1 \\ z_2 \end{bmatrix} \sim \mathcal{N}\left( \begin{bmatrix} a \cdot \mu \\ b \cdot \mu + c \end{bmatrix}, \Delta \begin{bmatrix} a \cdot a & a \cdot b \\ a \cdot b & b \cdot b \end{bmatrix} \right).$$

Using the law of iterated expectation, our average can be written as:

$$\langle a \cdot x \, \text{sign}(b \cdot x + c) \rangle = \mathbb{E}_{z_2}[\text{sign}(z_2)\mathbb{E}_{z_1|z_2}[z_1]]$$

$$= \mathbb{E}_{z_2}[\text{sign}(z_2)(a \cdot \mu + \frac{a \cdot b}{b \cdot b}(z_2 - (b \cdot \mu + c)))]$$

$$= (a \cdot \mu - \frac{a \cdot b}{b \cdot b}(b \cdot \mu + c))\mathbb{E}_{z_2}[\text{sign}(z_2)] + \frac{a \cdot b}{b \cdot b}\mathbb{E}_{z_2}[z_2\text{sign}(z_2)]$$

The first expectation follows from the definition of the cumulative distribution function $\Phi$

$$\mathbb{E}_{z_2}[\text{sign}(z_2)] = (1 - 2\Phi\left(\frac{-(b \cdot \mu + c)}{\sqrt{\Delta b \cdot b}}\right)).$$

The second term is simply the mean of a folded normal distribution

$$\mathbb{E}_{z_2}[z_2\text{sign}(z_2)] = (\sqrt{\Delta b \cdot b})\sqrt{\frac{2}{\pi}}\exp\left(\frac{-(b \cdot \mu + c)^2}{2\Delta b \cdot b}\right) + (b \cdot \mu + c)(1 - 2\Phi\left(\frac{-(b \cdot \mu + c)}{\sqrt{\Delta b \cdot b}}\right)).$$

Combining these three expressions we obtain the identity.

Integral 2:

$$\langle a \cdot x \, b \cdot x \rangle = (a \cdot \mu)(b \cdot \mu) + \Delta(a \cdot b) \tag{C.32}$$

where $x$ is defined as for the previous identity.

*Derivation.* We proceed as in the previous case. Define the auxiliary random variables $z_1 = a \cdot x$ and $z_2 = b \cdot x$. They follow a multivariate normal distribution

$$\begin{bmatrix} z_1 \\ z_2 \end{bmatrix} \sim \mathcal{N}\left( \begin{bmatrix} a \cdot \mu \\ b \cdot \mu \end{bmatrix}, \Delta \begin{bmatrix} a \cdot a & a \cdot b \\ a \cdot b & b \cdot b \end{bmatrix} \right).$$

Using the law of iterated expectation, our average may be written as:

$$\langle a \cdot x \, b \cdot x \rangle = \mathbb{E}_{z_2}[z_2\mathbb{E}_{z_1|z_2}[z_1]]$$

$$= \mathbb{E}_{z_2}[z_2(a \cdot \mu + \frac{a \cdot b}{b \cdot b}(z_2 - (b \cdot \mu)))]$$

$$= (a \cdot \mu - \frac{a \cdot b}{b \cdot b}(b \cdot \mu))\mathbb{E}_{z_2}[z_2] + \frac{a \cdot b}{b \cdot b}\mathbb{E}_{z_2}[z_2^2]$$

$$= (a \cdot \mu - \frac{a \cdot b}{b \cdot b}(b \cdot \mu))(b \cdot \mu) + \frac{a \cdot b}{b \cdot b}(\Delta b \cdot b + (b \cdot \mu)^2)$$

$$= (a \cdot \mu)(b \cdot \mu) + \Delta(a \cdot b).$$

## C.2   ODEs

We have now the building blocks to evaluate the expected values of Eqs. 7. We refresh the notation that $\delta^\mu = y^\mu - \hat{y}^\mu$, $y^\mu = \text{sign}\left(\boldsymbol{x}^\mu \cdot \overline{\boldsymbol{w}}_\mu/\sqrt{d}\right)$, and $\hat{y}^\mu = \boldsymbol{x}^\mu \cdot \boldsymbol{w}/\sqrt{d}$. Final step is to take the continuous limit. This is obtained by noticing that the RHS of the equations is factorised by $1/d$. Therefore by taking as time unit $1/d$ and defining time as $t = \mu/d$ the discrete updates converge to continuous increments as $d \to \infty$.

**Student-shift overlap $M$.**

$$\langle \Delta M \rangle = \frac{\eta}{d}\left(\rho v \alpha_+ + \rho M_+^* \beta_+ - (1 - \rho)v\alpha_- + (1 - \rho)M_-^* \beta_- - (M(v + \Delta^{mix}))\right) \tag{C.33}$$

*Derivation.* Starting from the definition in Eq. 7 for $M$

$$\langle \Delta M \rangle = \frac{\eta}{d}\left(\left\langle y\frac{\boldsymbol{x} \cdot \boldsymbol{v}}{\sqrt{d}}\right\rangle - \left\langle \frac{\boldsymbol{w} \cdot \boldsymbol{x}}{\sqrt{d}}\frac{\boldsymbol{x} \cdot \boldsymbol{v}}{\sqrt{d}}\right\rangle\right).$$

The first term can be evaluated using integral 1 and the second term using integral 2 yielding the result.

**Student-teacher $+$ overlap $R_+$.**

$$\langle \Delta R_+ \rangle = \frac{\eta}{d}\Big(\rho(M_+^*\alpha_+ + \beta_+) + (1-\rho)(-M_+^*\alpha_- + T_\pm\beta_-)$$
$$- \rho(MM_+^* + R_+\Delta_+) - (1-\rho)(MM_+^* + R_+\Delta_-)\Big) \tag{C.34}$$

*Derivation.*

$$\langle \Delta R_+ \rangle = \frac{\eta}{d}\left\langle \left(y - \frac{\boldsymbol{w}\cdot\boldsymbol{x}}{\sqrt{d}}\right)\left(\frac{\boldsymbol{x}\cdot\overline{\boldsymbol{w}}_+}{\sqrt{d}}\right)\right\rangle$$
$$= \frac{\eta}{d}\left(\rho\left\langle y\frac{\boldsymbol{x}\cdot\overline{\boldsymbol{w}}_+}{\sqrt{d}}\right\rangle_\oplus + (1-\rho)\left\langle y\frac{\boldsymbol{x}\cdot\overline{\boldsymbol{w}}_+}{\sqrt{d}}\right\rangle_\ominus - \rho\left\langle \frac{\boldsymbol{w}\cdot\boldsymbol{x}}{\sqrt{d}}\frac{\boldsymbol{x}\cdot\overline{\boldsymbol{w}}_+}{\sqrt{d}}\right\rangle_\oplus - (1-\rho)\left\langle \frac{\boldsymbol{w}\cdot\boldsymbol{x}}{\sqrt{d}}\frac{\boldsymbol{x}\cdot\overline{\boldsymbol{w}}_+}{\sqrt{d}}\right\rangle_\ominus\right).$$

These 4 terms can be computed using integrals 1 and 2 yielding the result.

**Student-teacher $-$ overlap $R_-$.**

$$\langle \Delta R_- \rangle = \frac{\eta}{d}\Big(\rho(M_-^*\alpha_+ + T_\pm\beta_+) + (1-\rho)(-M_-^*\alpha_- + \beta_-)$$
$$- \rho(MM_-^* + R_-\Delta_+) - (1-\rho)(MM_-^* + R_-\Delta_-)\Big) \tag{C.35}$$

*Derivation.* Same as for $R_+$.

**Self-overlap $Q$.**

$$\langle \Delta Q \rangle = \frac{2\eta}{d}\left(\rho(\alpha_+ M + \beta_+ R_+) + (1-\rho)(-\alpha_- M + \beta_- R_+) - M^2 - Q\Delta^{mix}\right)$$
$$+ \frac{\eta^2}{d}\Big(\Delta^{mix} + Q\Delta^{2mix} + M^2\Delta^{mix}$$
$$- 2\left(\rho\Delta_+(\alpha_+ M + \beta_+ R_+) + (1-\rho)\Delta_-(-\alpha_- M + \beta_- R_+)\right)\Big). \tag{C.36}$$

*Derivation.* This update requires additional steps with respect to the previous ones.

$$\langle \Delta Q \rangle = \frac{2\eta}{d}\left\langle \delta\frac{\boldsymbol{w}_j^\top\boldsymbol{x}}{\sqrt{d}}\right\rangle + \frac{\eta^2}{d}\left\langle (\delta^\mu)^2\frac{\|\boldsymbol{x}^\mu\|^2}{d}\right\rangle.$$

The first term is

$$\frac{2\eta}{d}\left\langle \delta\frac{\boldsymbol{w}_j^\top\boldsymbol{x}}{\sqrt{d}}\right\rangle = \frac{2\eta}{d}\left\langle y\frac{\boldsymbol{w}\cdot\boldsymbol{x}}{\sqrt{d}} - \left(\frac{\boldsymbol{w}\cdot\boldsymbol{x}}{\sqrt{d}}\right)^2\right\rangle$$
$$= \frac{2\eta}{d}\left(M(\rho\alpha_+ - (1-\rho)\alpha_-) + \rho\beta_+ R_+ + (1-\rho)\beta_- R_- - M^2 - Q\Delta^{mix}\right).$$

The second term

$$\frac{\eta^2}{d}\left\langle (\delta^\mu)^2\frac{\|\boldsymbol{x}^\mu\|^2}{d}\right\rangle = \frac{\eta^2}{d}\left\langle \left(y - \frac{\boldsymbol{w}\cdot\boldsymbol{x}}{\sqrt{d}}\right)^2\frac{\boldsymbol{x}\cdot\boldsymbol{x}}{d}\right\rangle$$
$$= \frac{\eta^2}{d}\left\langle y^2\frac{\boldsymbol{x}\cdot\boldsymbol{x}}{d} + \left(\frac{\boldsymbol{w}\cdot\boldsymbol{x}}{\sqrt{d}}\right)^2\frac{\boldsymbol{x}\cdot\boldsymbol{x}}{d} - 2y\frac{\boldsymbol{w}\cdot\boldsymbol{x}}{\sqrt{d}}\frac{\boldsymbol{x}\cdot\boldsymbol{x}}{d}\right\rangle$$

requires additional steps. We consider the three terms in the expression above, starting from the first one

$$\left\langle y^2\frac{\boldsymbol{x}\cdot\boldsymbol{x}}{d}\right\rangle = \left\langle \frac{\boldsymbol{x}\cdot\boldsymbol{x}}{d}\right\rangle = \frac{1}{d}(\sum_{i=1}^{d}\langle x_i^2\rangle) = \frac{1}{d}(\sum_{i=1}^{d}\rho\langle x_i^2\rangle_\oplus + (1-\rho)\langle x_i^2\rangle_\ominus)$$

$$= \frac{1}{d}\left(\sum_{i=1}^{d} \rho(\Delta_+ + v_i^2/d) + (1-\rho)(\Delta_- + v_i^2/d)\right) = \Delta^{mix} + v/d$$

$$= \Delta^{mix} + O(d^{-1}),$$

Where we used the simplification $y^2 = 1$ independently of the cluster's teacher. However, the remaining terms require us to split the expectation considering the probability of sampling from each cluster. The second term

$$\left\langle \frac{\boldsymbol{x} \cdot \boldsymbol{x}}{d}\left(\frac{\boldsymbol{w} \cdot \boldsymbol{x}}{\sqrt{d}}\right)^2\right\rangle = \rho\left\langle \frac{\boldsymbol{x} \cdot \boldsymbol{x}}{d}\left(\frac{\boldsymbol{w} \cdot \boldsymbol{x}}{\sqrt{d}}\right)^2\right\rangle_\oplus + (1-\rho)\left\langle \frac{\boldsymbol{x} \cdot \boldsymbol{x}}{d}\left(\frac{\boldsymbol{w} \cdot \boldsymbol{x}}{\sqrt{d}}\right)^2\right\rangle_\ominus.$$

We begin by analysing the average over the positive Gaussian and split $\boldsymbol{x}$ as $\boldsymbol{x} = \boldsymbol{v}/\sqrt{d} + \tilde{\boldsymbol{x}}$ such that $\tilde{\boldsymbol{x}}$ has zero mean. Then,

$$\left\langle \frac{\boldsymbol{x} \cdot \boldsymbol{x}}{d}\left(\frac{\boldsymbol{w} \cdot \boldsymbol{x}}{\sqrt{d}}\right)^2\right\rangle_\oplus = \left\langle \left[\frac{\boldsymbol{v} \cdot \boldsymbol{v}}{d^2} + \frac{2\boldsymbol{v} \cdot \tilde{\boldsymbol{x}}}{d\sqrt{d}} + \frac{\tilde{\boldsymbol{x}} \cdot \tilde{\boldsymbol{x}}}{d}\right]\left[\left(\frac{\boldsymbol{w} \cdot \boldsymbol{v}}{d}\right)^2 + 2\frac{\boldsymbol{w} \cdot \boldsymbol{v}}{d}\frac{\boldsymbol{w} \cdot \tilde{\boldsymbol{x}}}{\sqrt{d}} + \left(\frac{\boldsymbol{w} \cdot \tilde{\boldsymbol{x}}}{\sqrt{d}}\right)^2\right]\right\rangle_\oplus$$

Multiplying the terms in the brackets will give rise to 9 terms. We can see that the 3+3=6 terms corresponding to $\boldsymbol{v} \cdot \boldsymbol{v}/d^2$ and $2\boldsymbol{v} \cdot \tilde{\boldsymbol{x}}/d\sqrt{d}$ will tend to 0 in the limit of infinite d due to their scaling. We now analyse the other 3 terms:

Term 1:
$$\left\langle \frac{\tilde{\boldsymbol{x}} \cdot \tilde{\boldsymbol{x}}}{d}\left(\frac{\boldsymbol{w} \cdot \boldsymbol{v}}{d}\right)^2\right\rangle_\oplus = \left(\frac{\boldsymbol{w} \cdot \boldsymbol{v}}{d}\right)^2\left\langle \frac{\tilde{\boldsymbol{x}} \cdot \tilde{\boldsymbol{x}}}{d}\right\rangle_\oplus + O(d^{-1})$$
$$= M^2\Delta_+ + O(d^{-1}).$$

Term 2:
$$2\left\langle \frac{\tilde{\boldsymbol{x}} \cdot \tilde{\boldsymbol{x}}}{d}\frac{\boldsymbol{w} \cdot \boldsymbol{v}}{d}\frac{\boldsymbol{w} \cdot \tilde{\boldsymbol{x}}}{\sqrt{d}}\right\rangle_\oplus = 2R\left\langle \frac{\tilde{\boldsymbol{x}} \cdot \tilde{\boldsymbol{x}}}{d}\frac{\boldsymbol{w} \cdot \tilde{\boldsymbol{x}}}{\sqrt{d}}\right\rangle_\oplus$$
$$= 2R\left\langle \frac{\tilde{\boldsymbol{x}} \cdot \tilde{\boldsymbol{x}}}{d}\right\rangle_\oplus\left\langle \frac{\boldsymbol{w} \cdot \tilde{\boldsymbol{x}}}{\sqrt{d}}\right\rangle_\oplus + O(d^{-1})$$
$$= 0 + O(d^{-1}).$$

Term 3:
$$\left\langle \frac{\tilde{\boldsymbol{x}} \cdot \tilde{\boldsymbol{x}}}{d}\left(\frac{\boldsymbol{w} \cdot \tilde{\boldsymbol{x}}}{\sqrt{d}}\right)^2\right\rangle_\oplus = \left\langle \frac{\tilde{\boldsymbol{x}} \cdot \tilde{\boldsymbol{x}}}{d}\right\rangle_\oplus\left\langle \left(\frac{\boldsymbol{w} \cdot \tilde{\boldsymbol{x}}}{\sqrt{d}}\right)^2\right\rangle_\oplus + O(d^{-1})$$
$$= \Delta_+(\Delta_+ Q) + O(d^{-1}) = Q\Delta_+^2 + O(d^{-1}).$$

Thus finally,

$$\left\langle \frac{\boldsymbol{x} \cdot \boldsymbol{x}}{d}\left(\frac{\boldsymbol{w} \cdot \boldsymbol{x}}{\sqrt{d}}\right)^2\right\rangle = \rho(M^2\Delta_+ + Q\Delta_+^2) + (1-\rho)(M^2\Delta_- + Q\Delta_-^2)$$
$$= M^2\Delta^{mix} + Q\Delta^{2mix}.$$

For the the third term
$$\left\langle y\frac{\boldsymbol{w} \cdot \boldsymbol{x}}{\sqrt{d}}\frac{\boldsymbol{x} \cdot \boldsymbol{x}}{d}\right\rangle = \rho\left\langle y\frac{\boldsymbol{w} \cdot \boldsymbol{x}}{\sqrt{d}}\frac{\boldsymbol{x} \cdot \boldsymbol{x}}{d}\right\rangle_\oplus + (1-\rho)\left\langle y\frac{\boldsymbol{w} \cdot \boldsymbol{x}}{\sqrt{d}}\frac{\boldsymbol{x} \cdot \boldsymbol{x}}{d}\right\rangle_\ominus.$$

As before, we analyse the average over the positive Gaussian first and split $\boldsymbol{x}$ into its mean and a zero mean component:

$$\left\langle y\frac{\boldsymbol{x} \cdot \boldsymbol{x}}{d}\frac{\boldsymbol{w} \cdot \boldsymbol{x}}{\sqrt{d}}\right\rangle_\oplus = \left\langle y\left[\frac{\boldsymbol{v} \cdot \boldsymbol{v}}{d^2} + \frac{2\boldsymbol{v} \cdot \tilde{\boldsymbol{x}}}{d\sqrt{d}} + \frac{\tilde{\boldsymbol{x}} \cdot \tilde{\boldsymbol{x}}}{d}\right]\left[\frac{\boldsymbol{w} \cdot \boldsymbol{v}}{d} + \frac{\boldsymbol{w} \cdot \tilde{\boldsymbol{x}}}{\sqrt{d}}\right]\right\rangle_\oplus.$$

This gives rise to 6 terms. We can see that the 2+2=4 terms corresponding to $\boldsymbol{v} \cdot \boldsymbol{v}/d^2$ and $2\boldsymbol{v} \cdot \tilde{\boldsymbol{x}}/d\sqrt{d}$ will tend to 0 in the limit of infinite d due to their scaling. We now analyse the other 2 terms:

Term 1:

$$
\begin{aligned}
\left\langle y \frac{\tilde{\boldsymbol{x}} \cdot \tilde{\boldsymbol{x}}}{d} \frac{\boldsymbol{w} \cdot \boldsymbol{v}}{d} \right\rangle_{\oplus} &= M \left\langle y \frac{\tilde{\boldsymbol{x}} \cdot \tilde{\boldsymbol{x}}}{d} \right\rangle_{\oplus} \\
&= M \left\langle \operatorname{sign}(\frac{\tilde{\boldsymbol{x}} \cdot \overline{\boldsymbol{w}}_+}{\sqrt{d}} + \frac{\overline{\boldsymbol{w}}_+ \cdot \boldsymbol{v}}{d}) \frac{\tilde{\boldsymbol{x}} \cdot \tilde{\boldsymbol{x}}}{d} \right\rangle_{\oplus} \\
&= M \left\langle \operatorname{sign}(\frac{\tilde{\boldsymbol{x}} \cdot \overline{\boldsymbol{w}}_+}{\sqrt{d}} + \frac{\overline{\boldsymbol{w}}_+ \cdot \boldsymbol{v}}{d}) \right\rangle_{\oplus} \left\langle \frac{\tilde{\boldsymbol{x}} \cdot \tilde{\boldsymbol{x}}}{d} \right\rangle_{\oplus} + O(d^{-1}) \\
&= M \langle y \rangle_{\oplus} \Delta_+ + O(d^{-1}) \\
&= M \alpha_+ \Delta_+ + O(d^{-1}).
\end{aligned}
$$

Term 2:

$$
\begin{aligned}
\left\langle y \left( \frac{\tilde{\boldsymbol{x}} \cdot \tilde{\boldsymbol{x}}}{d} \right) \left( \frac{\boldsymbol{w} \cdot \tilde{\boldsymbol{x}}}{\sqrt{d}} \right) \right\rangle_{\oplus} &= \left\langle y \left( \frac{\boldsymbol{w} \cdot \tilde{\boldsymbol{x}}}{\sqrt{d}} \right) \right\rangle_{\oplus} \left\langle \left( \frac{\tilde{\boldsymbol{x}} \cdot \tilde{\boldsymbol{x}}}{d} \right) \right\rangle_{\oplus} + O(d^{-1}) \\
&= \Delta_+ \left\langle y \left( \frac{\boldsymbol{w} \cdot \tilde{\boldsymbol{x}}}{\sqrt{d}} \right) \right\rangle_{\oplus} + O(d^{-1}) \\
&= \Delta_+ R_+ \beta_+ + O(d^{-1}).
\end{aligned}
$$

Where the last equality follows using integral 1. Thus:

$$
\left\langle y \frac{\boldsymbol{x} \cdot \boldsymbol{x}}{d} \frac{\boldsymbol{w} \cdot \boldsymbol{x}}{\sqrt{d}} \right\rangle_{\oplus} = \Delta_+ (\alpha_+ M + \beta_+ R_+) + O(d^{-1}).
$$

We repeat the same analysis for the negative gaussian and get:

$$
\left\langle y \frac{\boldsymbol{x} \cdot \boldsymbol{x}}{d} \frac{\boldsymbol{w} \cdot \boldsymbol{x}}{\sqrt{d}} \right\rangle = \rho \Delta_+ (\alpha_+ M + \beta_+ R_+) + (1 - \rho) \Delta_- (-\alpha_- M + \beta_- R_+) + O(d^{-1}).
$$

Collecting everything together and taking the infinite dimensional limit:

$$
\langle \Delta \boldsymbol{w} \cdot \Delta \boldsymbol{w}/d \rangle = \frac{\eta^2}{d} \left( \Delta^{mix} + Q\Delta^{2mix} + M^2 \Delta^{mix} - 2 \left( \rho \Delta_+ (\alpha_+ M + \beta_+ R_+) + (1 - \rho)\Delta_-(-\alpha_- M + \beta_- R_+) \right) \right)
$$

Thus,

$$
\begin{aligned}
\langle \Delta Q \rangle = {}& \frac{2\eta}{d} \left( \rho(\alpha_+ M + \beta_+ R_+) + (1 - \rho)(-\alpha_- M + \beta_- R_+) - M^2 - Q\Delta^{mix} \right) \\
& + \frac{\eta^2}{d} \left( \Delta^{mix} + Q\Delta^{2mix} + M^2 \Delta^{mix} - 2 \left( \rho\Delta_+(\alpha_+ M + \beta_+ R_+) + (1 - \rho)\Delta_-(-\alpha_- M + \beta_- R_+) \right) \right).
\end{aligned}
$$

**Continuous limit.** Final step of the derivation is taking the termodynamics limit that leads to the ODEs implicitely defined in Thm. 3.3:

$$
\begin{aligned}
f_M(M, R_+, R_-, Q) = \eta \Big( & \rho v \alpha_+ + \rho M_+^* \beta_+ \\
& - (1 - \rho)v\alpha_- + (1 - \rho)M_-^* \beta_- - (M(v + \Delta^{mix})) \Big), \quad \text{(C.37)}
\end{aligned}
$$

$$
\begin{aligned}
f_{R_+}(M, R_+, R_-, Q) = \eta \Big( & \rho(M_+^* \alpha_+ + \beta_+) + (1 - \rho)(-M_+^* \alpha_- + T_\pm \beta_-) \\
& - \rho(MM_+^* + R_+ \Delta_+) - (1 - \rho)(MM_+^* + R_+ \Delta_-) \Big), \quad \text{(C.38)}
\end{aligned}
$$

$$
\begin{aligned}
f_{R_-}(M, R_+, R_-, Q) = \eta \Big( & \rho(M_-^* \alpha_+ + T_\pm \beta_+) + (1 - \rho)(-M_-^* \alpha_- + \beta_-) \\
& - \rho(MM_-^* + R_- \Delta_+) - (1 - \rho)(MM_-^* + R_- \Delta_-) \Big), \quad \text{(C.39)}
\end{aligned}
$$

$$f_Q(M, R_+, R_-, Q) = 2\eta \left( \rho(\alpha_+ M + \beta_+ R_+) + (1 - \rho)(-\alpha_- M + \beta_- R_+) - M^2 - Q\Delta^{mix} \right)$$
$$+ \eta^2 \Big( \Delta^{mix} + Q\Delta^{2mix} + M^2 \Delta^{mix} - 2\big(\rho\Delta_+(\alpha_+ M + \beta_+ R_+)$$
$$+ (1 - \rho)\Delta_-(-\alpha_- M + \beta_- R_+)\big)\Big).$$

(C.40)

# D   ODE solutions

In this section we first present the general solutions of the ODEs sketched in Theorem 3.4, then we specialise to the two scenarios discussed in the main text.

## D.1   General case

From the previous section, we have a system of coupled ODEs for the order parameters of the form:

$$\frac{dM}{dt} = c_1 + c_2 M,$$

$$\frac{dR_-}{dt} = c_{3-} + c_{4-} M + c_{5-} R_-,$$

$$\frac{dR_+}{dt} = c_{3+} + c_{4+} M + c_{5+} R_+,$$

$$\frac{dQ}{dt} = c_6 + c_7 M + c_8 M^2 + c_{9+} R_+ + c_{9-} R_- + c_{10} Q.$$

This represent a linear system of ODEs which can be solved using standard methods like Laplace transform, leading to Eqs. 10-12. We now report the equations including the exact expression of their coefficients.

$M$ :

$$M(t) = M_0 e^{-t\eta(v+\Delta^{mix})} + M_\infty(1 - e^{-t\eta(v+\Delta^{mix})}). \tag{D.41}$$

Where,

$$M_\infty = \frac{(\rho M_+^* \beta_+ + (1-\rho)M_-^* \beta_-) + v(\rho\alpha_+ - (1-\rho)\alpha_-)}{v + \Delta^{mix}}. \tag{D.42}$$

$R_+$:

$$R_+(t) = R_+^0 e^{-t\eta\Delta^{mix}} + R_+^\infty(1 - e^{-t\eta\Delta^{mix}}) + k_{1+}(e^{-t\eta\Delta^{mix}} - e^{-t\eta(v+\Delta^{mix})}). \tag{D.43}$$

Where,

$$R_+^\infty = \frac{(\rho\beta_+ + T_\pm(1-\rho)\beta_-) + M_+^*(\rho\alpha_+ - (1-\rho)\alpha_- - M_\infty)}{\Delta^{mix}}, \tag{D.44}$$

$$k_{1+} = \frac{M_+^*(M_\infty - M_0)}{v}. \tag{D.45}$$

$R_-$:

$$R_-(t) = R_-^0 e^{-t\eta\Delta^{mix}} + R_-^\infty(1 - e^{-t\eta\Delta^{mix}}) + k_{1-}(e^{-t\eta\Delta^{mix}} - e^{-t\eta(v+\Delta^{mix})}). \tag{D.46}$$

Where,

$$R_-^\infty = \frac{(T_\pm\rho\beta_+ + (1-\rho)\beta_-) + M_-^*(\rho\alpha_+ - (1-\rho)\alpha_- - M_\infty)}{\Delta^{mix}}, \tag{D.47}$$

$$k_{1-} = \frac{M_-^*(M_\infty - M_0)}{v}. \tag{D.48}$$

$Q$:

$$\begin{aligned}
Q(t) = & Q_0 e^{-t\eta(2\Delta^{mix} - \eta\Delta^{2mix})} + Q_\infty(1 - e^{-t\eta(2\Delta^{mix} - \eta\Delta^{2mix})}) \\
& + k_2(e^{-t\eta(2\Delta^{mix} - \eta\Delta^{2mix})} - e^{-t\eta\Delta^{mix}}) \\
& + k_3(e^{-t\eta(2\Delta^{mix} - \eta\Delta^{2mix})} - e^{-t\eta(v+\Delta^{mix})}) \\
& + k_4(e^{-t\eta(2\Delta^{mix} - \eta\Delta^{2mix})} - e^{-t\eta(2v+2\Delta^{mix})}).
\end{aligned} \tag{D.49}$$

Where,

$$Q_\infty = \frac{\eta\Delta^{mix} + 2\rho\beta_+ R_+^\infty(1 - \eta\Delta_+) + 2(1-\rho)\beta_- R_-^\infty(1 - \eta\Delta_-)}{2\Delta^{mix} - \eta\Delta^{2mix}},$$

$$+ \frac{M_\infty(M_\infty(\eta\Delta^{mix} - 2) + 2\rho\alpha_+(1 - \eta\Delta_+) - 2(1 - \rho)\alpha_-(1 - \eta\Delta_-))}{2\Delta^{mix} - \eta\Delta^{2mix}}, \qquad \text{(D.50)}$$

$$k_2 = \frac{2\rho\beta_+(1 - \eta\Delta_+)(R_+^\infty - R_+^0 - k_{1+}) + 2(1 - \rho)\beta_-(1 - \eta\Delta_-)(R_-^\infty - R_-^0 - k_{1-})}{\Delta^{mix} - \eta\Delta^{2mix}}, \quad \text{(D.51)}$$

$$k_3 = \frac{2\rho\beta_+(1 - \eta\Delta_+)k_{1+} + 2(1 - \rho)\beta_-(1 - \eta\Delta_-)k_{1-}}{\Delta^{mix} - \eta\Delta^{2mix} + v},$$

$$+ \frac{(M_\infty - M_0)(M_\infty(\eta\Delta^{mix} - 2) + 2\rho\alpha_+(1 - \eta\Delta_+) - 2(1 - \rho)\alpha_-(1 - \eta\Delta_-))}{\Delta^{mix} - \eta\Delta^{2mix} + v},$$
$$\text{(D.52)}$$

$$k_4 = \frac{(\eta\Delta^{mix} - 2)(M_\infty - M_0)^2}{\eta\Delta^{2mix} + 2v}. \qquad \text{(D.53)}$$

### D.2  Spurious correlations setting

Under the setting discussed in the Sec. 4.1 ($\rho = 0.5, \Delta_+ = \Delta_- = \Delta, T_\pm = 1$), we can make the following simplifications:

1. $\Delta^{mix} = \Delta$,
2. $\Delta^{2mix} = \Delta^2$,
3. $\alpha_+ = -\alpha_- = \alpha$,
4. $\beta_+ = \beta_- = \beta$,
5. $M_+^* = M_-^* = M^*$,
6. $R_+ = R_- = R$.

The equations then take the form:

$$M(t) = M_0 e^{-t\eta(v+\Delta)} + M_\infty(1 - e^{-t\eta(v+\Delta)}),$$
$$R(t) = R^0 e^{-t\eta\Delta} + R^\infty(1 - e^{-t\eta\Delta}) + k_1(e^{-t\eta\Delta} - e^{-t\eta(v+\Delta)}),$$
$$Q(t) = Q_0 e^{-t\eta(2\Delta - \eta\Delta^2)} + Q_\infty(1 - e^{-t\eta(2\Delta - \eta\Delta^2)})$$
$$+ k_2(e^{-t\eta(2\Delta - \eta\Delta^2)} - e^{-t\eta\Delta})$$
$$+ k_3(e^{-t\eta(2\Delta - \eta\Delta^2)} - e^{-t\eta(v+\Delta)})$$
$$+ k_4(e^{-t\eta(2\Delta - \eta\Delta^2)} - e^{-t\eta(2v+2\Delta)}).$$

Where,

$$M_\infty = \frac{M^*\beta + v\alpha}{v + \Delta},$$
$$R_\infty = \frac{\beta + M^*(\alpha - M_\infty)}{\Delta},$$
$$k_1 = \frac{(M_\infty - M_0)}{v},$$
$$Q_\infty = \frac{\eta\Delta + 2\beta R_\infty(1 - \eta\Delta)}{2\Delta - \eta\Delta^2} + \frac{M_\infty(M_\infty(\eta\Delta - 2) + 2\alpha(1 - \eta\Delta))}{2\Delta - \eta\Delta^2},$$
$$k_2 = \frac{2\beta(1 - \eta\Delta)(R_\infty - R_0 - k_1)}{\Delta - \eta\Delta^2},$$
$$k_3 = \frac{2\beta(1 - \eta\Delta)k_1}{\Delta - \eta\Delta^2 + v} + \frac{(M_\infty - M_0)(M_\infty(\eta\Delta - 2) + 2\alpha(1 - \eta\Delta))}{\Delta - \eta\Delta^2 + v},$$
$$k_4 = \frac{(\eta\Delta - 2)(M_\infty - M_0)^2}{\eta\Delta^2 + 2v}.$$

### D.3 Fairness setting

The general fairness case coincides with the general case discussed above (D.1), therefore we limit our discussion to the simplified case with centered clusters.

Under the zero shift $v = 0$, the equations take the simplified form wherein $M, v, M_\pm^*$ are 0, the transient term in $R_\pm$ vanishes and $Q$ only has one transient term. Specifically:

$$R_+(t) = R_+^0 e^{-t\eta\Delta^{mix}} + R_+^\infty(1 - e^{-t\eta\Delta^{mix}}),$$

$$R_-(t) = R_-^0 e^{-t\eta\Delta^{mix}} + R_-^\infty(1 - e^{-t\eta\Delta^{mix}}),$$

$$Q(t) = Q_0 e^{-t\eta(2\Delta^{mix} - \eta\Delta^{2mix})} + Q_\infty(1 - e^{-t\eta(2\Delta^{mix} - \eta\Delta^{2mix})}) + Q_{trans}(e^{-t\eta(2\Delta^{mix} - \eta\Delta^{2mix})} - e^{-t\eta\Delta^{mix}}).$$

Where

$$R_+^\infty = \sqrt{\frac{2}{\pi}} \frac{\rho\sqrt{\Delta_+} + T_\pm(1-\rho)\sqrt{\Delta_-}}{\Delta^{mix}},$$

$$R_-^\infty = \sqrt{\frac{2}{\pi}} \frac{T_\pm\rho\sqrt{\Delta_+} + (1-\rho)\sqrt{\Delta_-}}{\Delta^{mix}},$$

$$Q_\infty = \frac{\eta\Delta^{mix} + 2\sqrt{\frac{2}{\pi}}\rho\sqrt{\Delta_+}R_+^\infty(1 - \eta\Delta_+) + 2\sqrt{\frac{2}{\pi}}(1-\rho)\sqrt{\Delta_-}R_-^\infty(1 - \eta\Delta_-)}{2\Delta^{mix} - \eta\Delta^{2mix}},$$

$$Q_{trans} = \sqrt{\frac{2}{\pi}} \frac{2\rho\sqrt{\Delta_+}(1 - \eta\Delta_+)(R_+^\infty - R_+^0) + 2(1-\rho)\sqrt{\Delta_-}(1 - \eta\Delta_-)(R_-^\infty - R_-^0)}{\Delta^{mix} - \eta\Delta^{2mix}}.$$

## E  Deeper analysis of the learning dynamics equations

This section provides insights into the learning dynamics — particularly those relevant to bias evolution — that arise out of the expressions for order parameter evolution. We shall provide intuitive explanations behind the various mathematical terms that appear.

### E.1  Single centered cluster

Consider first a single cluster centered at the origin–i.e. $\rho = 1, v = 0$ with variance $\Delta$. In this setting, the minimum generalisation error is achieved when the student perfectly aligns with the teacher and optimises its norm such that $Q_{opt} = \frac{2}{\pi\Delta}$, achieving the generalisation error $\epsilon_{\min} = 1 - \frac{2}{\pi}$.

Importantly, this is not 0 since the student and the teacher are mismatched –i.e. the student is linear whereas the teacher has a $sign(\cdot)$ activation function. From the equations, we observe that the asymptotic generalisation error when training using online stochastic gradient descent in this setting is

$$\epsilon_\infty = \frac{1 - 2/\pi}{1 - \eta\Delta/2} = \left(1 - \frac{2}{\pi}\right)\left(1 + \frac{\eta\Delta}{2} + O(\eta^2\Delta^2)\right). \tag{E.54}$$

Thus, as the learning rate increases, the generalisation error increases until it reaches the critical learning rate beyond which training is unstable and the loss grows unboundedly. In the single cluster case, Eq. E.54 this is $2/\Delta$ which matches the classical result from convex optimisation [23]. We can similarly find the critical learning rate for two clusters to be $2\Delta^{mix}/\Delta^{2mix}$ by ensuring exponential terms decay to zero in equation 12.

### E.2  Analysis of teacher alignment ($\tau_R$) and student magnitude ($\tau_Q$) timescales

We now consider the fairness setting with zero shift as illustrated in Fig. 1c. As discussed in section 4.2, the relevant timescales in this setting are

$$\tau_R = \frac{1}{\eta\Delta^{mix}}, \qquad \tau_Q = \frac{1}{\eta(2\Delta^{mix} - \eta\Delta^{2mix})},$$

since $M(t)$ is always zero. Fig. 7 shows the crossing phenomena of the loss curves along with the order parameter evolution and other insightful terms. The alignment of the student is governed by the

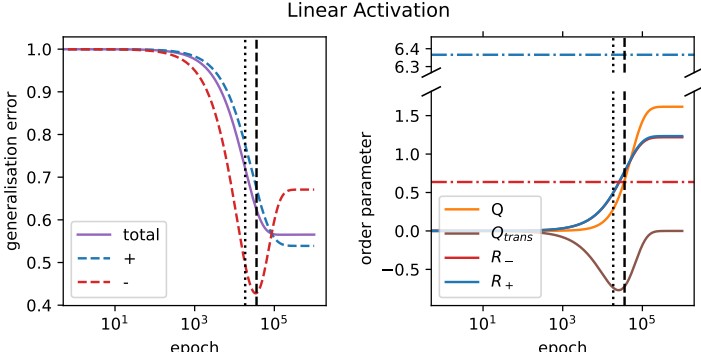

Figure 7: **The Crossing Phenomenon** The left shows the 'crossing' of the loss curves on the negative sub-population in red (higher variance and lower representation) and positive sub-population in blue (lower variance but greater representation) along with the overall loss in purple obtained as a weighted average of the two. It also marks $\tau_R$ as the dashed vertical line and $\tau_Q$ as the dotted vertical line. The right side shows the evolution of the order parameters and a transient term. The horizontal blue and red dash-dotted line mark the optimal value of Q for the positive-subpopulation and negative sub-populations respectively. The parameters are $v = 0, \rho = 0.8, \Delta_+ = 0.1, \Delta_- = 1, T_\pm = 0.9, \eta = 0.1$.

timescale $\tau_R$ and the change in its magnitude is governed by the timescale $\tau_Q$. Initially, the classifier has a small magnitude and its alignment roughly matches the two teachers which are themselves quite similar ($T_\pm = 0.9$). Indeed, we see that the $R_+$ and $R_-$ have very similar trajectories. However, smaller magnitudes advantage higher variances as discussed in Appendix E.1 ($Q_{opt}$ is inversely proportional to the cluster variance).

We mark the optimal values of $Q$ using horizontal lines in Fig.7 on the left side with blue for the positive sub-population (lower variance) and red for the negative sub-population (higher variance). As the magnitude of the student grows, we observe a sharp drop in the generalisation error on the higher variance sub-population till $Q$ crosses the horizontal red line. Beyond this point, the generalisation error on the higher variance sub-population rises since the magnitude of the student has exceeded the optimal value (horizontal red line) and the generalisation error on the lower variance sub-population continues to fall as the magnitude of the student approaches the horizontal blue line. Finally, an inspection of the timescales reveals that $\tau_Q$ (vertical dotted line) is less than $t_R$ (vertical dashed line) and hence we may expect the student magnitude to saturate before its alignment. However, $Q_{trans}$, the transient term associated with $Q$ (third line of equation 12), is always negative and hence suppresses the growth of $Q$ initially.

In summary, we observe a two phase behaviour. First the student shifts its alignment and increases magnitude leading to a sharper drop in the higher variance generalisation error. Second, we observe that as the student continues increasing magnitude while keeping its alignment fixed, it advantages the lower variance cluster.

### E.3  Asymptotic preference

This section discusses the asymptotic generalisation errors of our classifier when $v = 0$ as a function of representation and variances. Firstly, as discussed in section 4.2,

$$R_+^\infty > R_-^\infty \iff \rho\sqrt{\Delta_+} > (1 - \rho)\sqrt{\Delta_-}.$$

Intuitively, one might expect that the asymptotically lower generalisation error is achieved on the population whose teacher has better asymptotic alignment with the student. Indeed, when the learning rate tends to 0, we observe exactly this as illustrated by the two dark phases in Fig. 8 on the left side. However, when the learning rate is greater than zero, we observe more complex behaviour. Fig. 8 *(right)* shows the emergence two new phases (light red and light blue) wherein the classifier exhibits higher generalisation error on a sub-population despite having better alignment with its corresponding teacher. This behaviour can be traced back to equation E.54 wherein the increase in asymptotic generalisation error due to non-zero learning rates is amplified by the cluster variance.

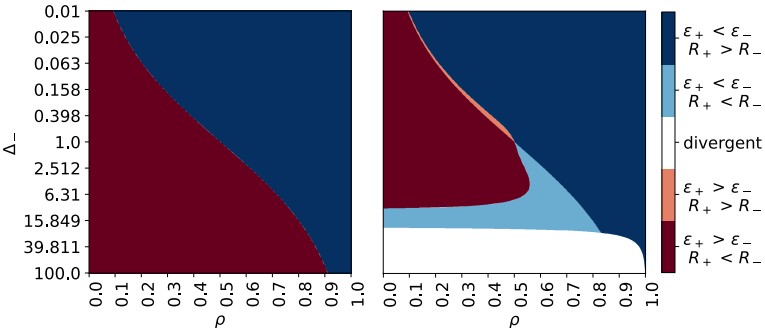

Figure 8: **Initial and Asymptotic student preferences** We set $v = 0, \Delta_+ = 1, T_\pm = 0.9, \eta = 0.1$ and study the values of $\rho, \Delta_-$. The figure studies only asymptotic preferences under $v = 0, \Delta_+ = 1, T_\pm = 0.9$. When the learning rate is small ($\eta \to 0^+$ on *left side*), the cluster which has better alignment with the teacher must also have lower generalisation error. However, for non-zero learning rates ($\eta = 0.1$ on *right side*), behaviour is more complicated leading to the light colored phases where despite better asymptotic alignment with the teacher, the generalisation error is higher. Parameters: $\eta \to 0^+$ (left) vs $\eta = 0.1$ (right).

Thus, our analysis shows how a large learning rate can also become a source of bias in our classifier by advantaging the sub-population with smaller variance.

# F Additional numerical simulations

## F.1 CelebA

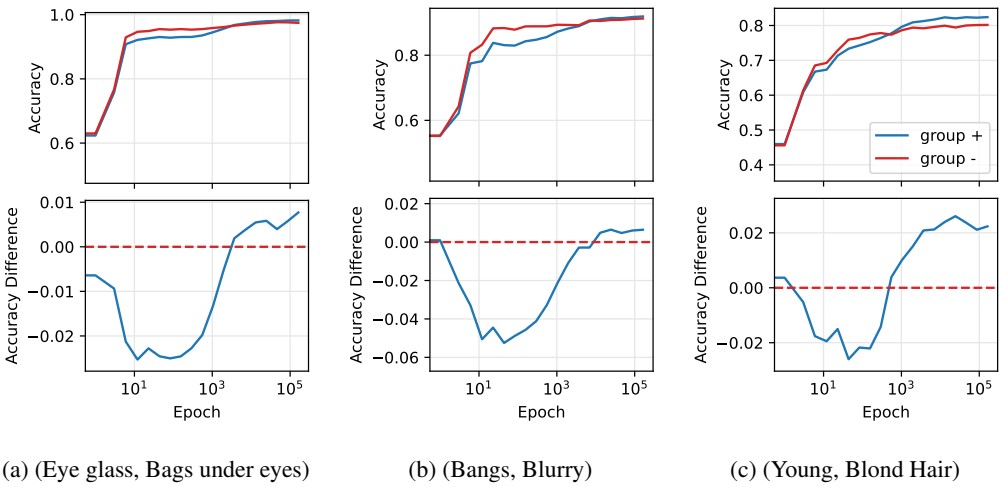

(a) (Eye glass, Bags under eyes)     (b) (Bangs, Blurry)     (c) (Young, Blond Hair)

Figure 9: **Numerical simulations in the CelebA dataset.** Figure shows the average accuracy (solid lines) and standard deviation (shaded area) of 4 different runs in this framework. The top row depicts the test accuracy over the course of training for different pairs of target and group attributes. The bottom row illustrates the difference in test accuracies between the $+$ and $-$ subpopulations, highlighting the crossing phenomenon observed during training. *Panels (a), (b)*, and *(c)* depict this for the pairs of target and group attributes of (Eye glass, Bags under eyes), (Bangs, Blurry), and (Young, Blond Hair), respectively.

The goal of this experiment is to show the emergence of different timescales in realist scenarios of relevance for the fairness literature.

The CelebA dataset [27] contains over 200k celebrity images annotated with 40 attribute labels, covering a wide range of facial attributes such as gender, age, and expressions. For this experiment, we consider different pairs as the target and group attributes. The task is to predict the target attribute while the group attribute defines the $+$ and $-$ subpopulations.

For the model, we select a pretrained ResNet-18 model on ImageNet and add an additional fully connected layer, with only the latter being optimised during training. We use cross-entropy as the loss objective and train via online SGD.

We randomly selected target-label pairs, making sure to avoid attributes that are pathologically underrepresented in the dataset and would hinder the significance of the result. In the plots shown in Fig. 9 we show some of the pairs that show a crossing phenomenon. Each panel in Fig. 9 show the accuracy and accuracy gap over the course of training. Notice how the classifier favours sub-population $-$ in the initial phase of training before changing preference.

This result shows that bias can change over the course of training even in standard setting. This does not imply that it will always occur and indeed several of the pairs in the dataset do not show a crossing phenomenon. However, understanding when and why this phenomenon occurs can affect the algorithmic choices that we make in our ML pipeline.

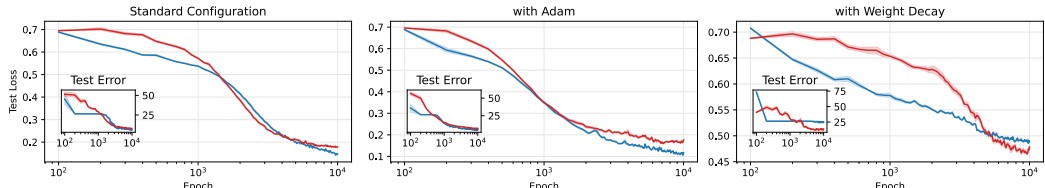

Figure 10: **Synthetic Data Simulation with alternate Training Protocols** We observe the 'double-crossing' phenomena in not only the loss curves, but also the error curves for the positive sub-population (blue) and the negative sub-population (red) *(left)*. The shaded areas quantify the standard deviation obtained across 10 seeds. We observe similar behavior when using Adam *(middle)* and weight decay *(right)*. The data distribution parameters are $d = 100, v = 4, \rho = 0.7, \Delta_+ = 0.1, \Delta_- = 1, T_\pm = 0.9, \eta = 0.01, \alpha_+ = 0.471, \alpha_- = -0.188$

In this section we test the validity of the prediction of our model across various networks and training protocols. We consider a data distribution with parameters as detailed in the caption of Fig. 10. We train a 2-layer MLP with ReLU activation in the hidden layer(s) and sigmoidal activation at the output with 128 neurons in the hidden layer with online SGD using BCE loss. We refer to this as the 'standard configuration' for further comparisons. We sample training and test data from the data distribution and use the test data to obtain estimates of the loss as well as error rates (percentage of test examples misclassified). We visualize these in Fig. 10 on the left.

We observe the three phase behaviour predicted by our model. The positive sub-population is initially advantaged more since it exhibits stronger spurious correlation. Then, the negative sub-population is advantaged since it has a higher variance. Finally, as per Eq. 16, the positive-sub-population is advantaged once more since it has sufficiently high representation. We not only observe the 'double-crossing' phenomena in the losses, but also in the test errors demonstrating the robustness of our model beyond the linearity and MSE loss assumptions.

We also observe similar behavior using Adam optimization in Fig. 10 (center). We note that we had to use a smaller learning rate of 0.001 to keep training stable since Adam leads to faster optimization manifesting in a higher effective learning rate than SGD.

When we train using a weight decay penalty of 0.1 (Fig. 10 right), we observe that asymptotically the higher variance cluster is now preferred. This behaviour can be explained using the theoretical model. As discussed in Appendix E, smaller student magnitude advantages the higher variance group and weight decay encourages smaller student weights leading to an asymptotic preference for the negative sub-population.

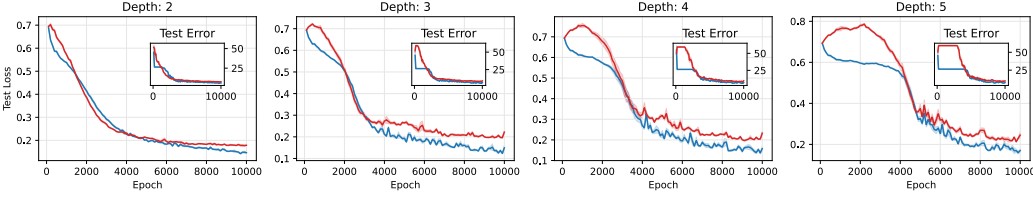

Figure 11: **Ablations with Deeper Networks** We observe the 'double-crossing' phenomena across deeper networks as well.

We now train deeper networks with 2 to 5 layers and visualise results in Fig. 11. The 'double-crossing' phenomena persists across deeper networks. We also train wider networks with 2, 16, 128 and 1024 units in the hidden layer and visualise results in Fig. 12. The 'double-crossing' phenomena persists

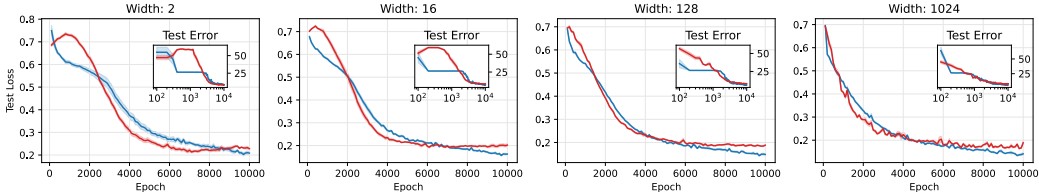

Figure 12: **Ablations with Wider Networks** We observe the 'double-crossing' phenomena across wider networks as well.

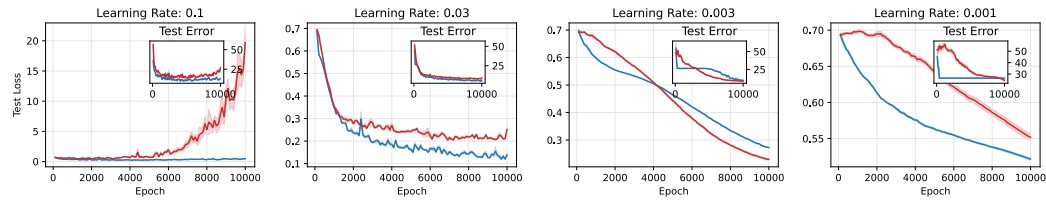

Figure 13: **Ablations across Learning Rates** Larger learning rates can lead to instability *(left)*. If training is stable however, we observe the 'crossing' phenomena as usual, just at different time scales due to different speeds of training.

across wider networks.

Finally, we vary the learning rate as well from our standard configuration and note that when the learning rate is too high, the training is unstable for the higher variance cluster (Fig. 13 left). For stable training, however, we observe the 'double-crossing' phenomena as usual, just at longer timescales for slower learning rates as predicted by our theory.

