# OpenReview forum: "Bias in Motion: Theoretical Insights into the Dynamics of Bias in SGD Training"
_NeurIPS.cc/2024/Conference — NeurIPS 2024 poster_

### Official Review · Reviewer_43yt · 2024-07-05

**Soundness:** 3
**Presentation:** 3
**Contribution:** 2
**Rating:** 6
**Confidence:** 3

**Summary:**

This paper presents a theoretical analysis of bias dynamics in machine learning models during the training process.

1. The authors introduce a high-dimensional teacher-student framework called the Teacher-Mixture (TM) model to study how bias emerges and evolves during stochastic gradient descent (SGD) training of linear classifiers.

2. They provide an analytical solution for the dynamics of key order parameters that characterize the classifier's performance and bias in the high-dimensional limit.

3. The analysis reveals a rich three-phase learning process where the classifier exhibits biases toward different data features at different stages of training:
   - Initial phase: Influenced by sub-populations with strong class imbalance
   - Intermediate phase: Dominated by the saliency (norm) of samples in sub-populations
   - Final phase: Determined by sub-population imbalance (relative representation)

4. The authors show that bias can evolve non-monotonically during training, potentially leading to "crossing" phenomena where the preferred sub-population changes over time.

5. They validate their theoretical findings through numerical experiments on both synthetic data and real datasets (CIFAR10, MNIST, CelebA), demonstrating that the observed phenomena extend beyond the simplifying assumptions of their model.

6. The paper discusses implications for fairness and robustness in machine learning, highlighting how different properties of sub-populations can generate and amplify bias at different timescales.

7. The authors argue that understanding these transient dynamics is crucial for developing effective bias mitigation strategies, especially under limited computational resources.

**Strengths:**

this work provides a theoretical foundation for understanding the complex time-dependent nature of bias in machine learning models during training, which was previously, as claimed by the authors (I can only judge this superficially) not well captured by existing theoretical studies focused on asymptotic analysis.​​​​​​​​​​​​​​​​
As far as I can judge the paper is well written and the experiments, results well described.

**Weaknesses:**

I can not really judge the novelty and soundness of the proposed work but I believe the paper is of high quality. Of course the setting considered is far from „large scale“ but I dont see this as too important at this stage.

**Questions:**

Can you link training on MNIST, etc closer to the theory by training again on linear models. Can you make the jump between theoretical and experimental section less severe by e.g. studying linear models trained on different (more difficult and not theoretically motivated) data generation processes? Can you provide examples for linear models where we do not observe crossing phenomena to build up more intuition?

**Limitations:**

Yes

---

> ### Author Rebuttal · Authors · 2024-08-07
>
> We would like to thank the reviewer for their time reading and assessing our paper. We are happy that the reviewer found our paper well-written and, in the following, we would like to address their main comments touching an important point, i.e. the robustness of our result in more complex settings.
>
> We extended the experiment on MNIST by now training on a network with no ReLU non-linearities –i.e. linear activations– and visualise the result in Figure 2 of the rebuttal PDF. Compared with Figure 5(a) of the paper, we note that removing the non-linearities leads to similar qualitative behaviour and we still observe a crossing. Giving another confirmation of our theory.
>
> The reviewer also asked about situations where the crossing does not occur. We can find an instance in Fig.1a of the main paper and, more broadly, looking at the phase diagram of Fig.3b we can notice that all the dark red and dark blue regions of the generative model’s parameters are instances where the crossing does not occur. Our theory shows that this is because the preferred subpopulation of our classifier stays the same across the different phases of dynamics described in the paper.
>
> Connected to this point, we would like to refer to answer to reviewer CSav where we also see that the crossing is not necessarily happening –Fig.1 and Fig.3 of the rebuttal PDF– and it occurs when the data distributions of the groups show characteristics identified by our theory –that depends on the variance and relative representation of the groups as explained in section 4.2 of the paper. We kindly ask the reviewer to refer to CSav rebuttal for further details.
>
> We hope we were able to address your concern and once again thank you for your valuable time and remarks.

---

> ### Author Response · Authors · 2024-08-13
>
> We hope that the clarifications and additional experiments have addressed the reviewer’s  concerns. If there are any remaining questions or points that need further discussion, we would be grateful for any additional feedback.

---

### Official Review · Reviewer_a8nr · 2024-07-09

**Soundness:** 3
**Presentation:** 3
**Contribution:** 3
**Rating:** 7
**Confidence:** 2

**Summary:**

The manuscript tries to understand the training dynamics in SGD training, through the lens of synthetic data with sub-populations from the Gaussian-mixture model. The paper leverages high-dimensional analysis to prove that the solution converges to a set of ODEs and thus characterizes the evolution of bias throughout training, namely a three-phase learning process. The study provides a unifying view of learning in fairness and spurious correlation problems, highlighting the presence of ephemeral biases characterized by multiple timescales during training. These theoretical insights are further validated on practical datasets, like CIFAR, MNIST, and CelebA.

In general, the take-away messages of the manuscript look interesting.

**Strengths:**

* The manuscript is well-structured, with a clear logical flow.
* The idea of using a theoretical model, namely a linear model with a mixture of Gaussian distribution, followed by using order parameters to depict the generalization error, is quite interesting.
* The manuscript leverages the theory model to interpret the training dynamics when spurious correlations and bias are present during the training.

**Weaknesses:**

The reviewer cannot spot significant weaknesses in the manuscript.

**Questions:**

* How does the value of $m$ affect Theorem 3.4 and the insights discussed in Section 4?

---

> ### Author Rebuttal · Authors · 2024-08-07
>
> We would like to thank the reviewer for their general appreciation of our work and for their comment.
>
> **Impact of the value of $m$**
>
>
> Thank you for your insightful question. We plan to incorporate the following points into the revised manuscript:
>
> * A higher value of $m$ would introduce additional order parameters for the student-teacher overlaps and student-shift overlaps corresponding to the extra clusters. Despite this added complexity,  we anticipate a similar system of coupled ODEs to what is described in Appendix D.1, and we are confident that the results would remain analytically tractable.
>
> * We expect that the form of the equations in Theorem 3.4 would remain fundamentally unchanged, with the same type of timescales and dynamics discussed in the paper. The constants, such as $\Delta^{mix}$,  would be adjusted to account for the weighted average across a greater number of clusters. Consequently, the same factors to drive bias –degree of spurious correlation, variance, relative representation– and the three-phase behaviour described in the paper would still emerge.  We are optimistic that the key takeaways from Section 4 will extend to the $m>2$ setting.
>
> * While we believe most conclusions will hold for higher values or $m$, , this is an interesting direction we are currently investigating. In this work, we focused on  $m=2$,  which already shows a rich and sometimes counter-intuitive phenomenology.    However, the $m>2$ scenario introduces additional complexities, such as  varying levels of similarity between groups, which merit further exploration.
>
> We would welcome any further suggestions or discussion to address this concern more comprehensively.

---

> > ### Comment · Reviewer_a8nr · 2024-08-08
> > **Thank you for the rebuttal**
> >
> > The reviewer appreciates the authors for providing the rebuttal and will maintain the original score.

---

> > > ### Author Response · Authors · 2024-08-08
> > >
> > > Thank you for your appreciation. Is anything we can add to improve rating and confidence scores?

---

### Official Review · Reviewer_CSav · 2024-07-17

**Soundness:** 3
**Presentation:** 3
**Contribution:** 3
**Rating:** 6
**Confidence:** 2

**Summary:**

The paper analyzes how different subgroups of the data affect optimization dynamics. The analysis is done in a large-dimension limit by analyzing squared error of a linear student classifier modeling a piece-wise linear teacher one. This analysis provides insights into dynamic loss behavior of loss on different subgroups and which aspects of the data lead cause this.

**Strengths:**

- the paper provides a different analysis of SGD-optimization.
- predicts a double-cross phenomenon where different stages of training
- shows evidence of those double-cross phenomenon in real experiments.

**Weaknesses:**

- the student is underpowered in this setup. The teacher is piece-wise linear but the student is purely linear. This fundamentally imposes a trade-off between modeling one subset of the data better than another. This breaks out of the case where models we use these days can perfectly fit any training dataset even with medium-sized datasets like CelebA.

Some missing related work: https://proceedings.mlr.press/v162/pezeshki22a.html, https://arxiv.org/abs/2308.12553

- This is important: What were the hyperparameters used in the various experiments? What happens as you vary each of them (like learning rate and weight decay)? It seems to me that the synthetic model cannot account for any of these parameters, so it feels like there is a gap in what the theory actually explains (especially with the underpowered student).

- (Minor) Adam is the more common optimizer and it is unclear whether the theory in the paper extends to this and if the insights about features translate. Can the authors comment on this?

**Questions:**

Is there a way you can modify the data distribution in the real experiments according to what your theory suggests lead to certain behavior, and show that the behavior is now different? This would seal the theory and main reason behind the seen curves.

otherwise, See weaknesses.

**Limitations:**

See weaknesses.

---

> ### Author Rebuttal · Authors · 2024-08-07
>
> We thank the reviewer for their appreciation of our analysis and their comments. We feel that these are important comments to improve the clarity of our work and we thank the reviewer for pointing them out, we ran additional experiments (discussed below) to address them.
>
> **Limitations of the model**
>
> Our theoretical analysis is limited to purely linear classifiers to maintain analytical tractability. To address questions about the robustness of the results and whether this may be due to the low capacity of the classifier, we relied on numerical evaluation on deeper networks. We provided some experiments in section 5 and appendix F where we used 2 layer MLPs and finetuned a ResNet-18 on real data sets.
>
> Taking your remark and the related remark of reviewer JTC4 into account, we have now included additional results that demonstrate the behaviour predicted by our theory on deeper and wider MLPs (see the submitted pdf in our global response):
>
> - We have expanded the experiment in section F.2 as follows: We consider a data distribution with $d=100$, $\rho=0.7$, $\Delta_+=0.1$, $\Delta_-=1$, $T_{\pm}=0.9$, $v=4$, $\alpha_+ = 0.471$,  $\alpha_- = -0.188$. We train a 2-layer MLP with ReLU activation in the hidden layer(s) and sigmoidal activation at the output with 128 neurons in the hidden layer with online SGD with a learning rate of 0.01 using BCE loss. We refer to this as the ‘standard configuration’ useful for further comparisons. We visualise the loss curves and test error rates on the positive (blue)  and negative (red) subpopulations along with standard deviation across 10 seeds in Figure 4 (left panel).
>
> - We further train MLPs of increasing depth (2,3,4,5 layers) with 128 hidden neurons in each layer and visualise the loss and error curves in Figure 6. The figures show the three phase behaviour predicted by our theory - the positive subpopulation (in blue) is learnt faster initially since it exhibits stronger spurious correlation with the shift vector; the negative subpopulation (in red) is learnt faster subsequently due to higher variance and finally the positive subpopulation is asymptotically prioritised once more due to higher relative representation. As such, we also observe the ‘double-crossing’ phenomena discussed in the paper. We stress that this result and the next are obtained by taking exactly the same parameters used in Fig.10 of the paper without exploring the phase space any further.
>
> - We also train MLPs of 2 layers but increasing width (2, 16, 128 and 1024 hidden neurons) and visualise loss and errors curves in Figure 7. We note the presence of the same three phase behaviour across increasing parameterization. Thus, our empirics with over-parameterized students show qualitatively similar behaviour. We hope this further empirical validation helps show the validity of our theory in settings closer to modern regimes.
>
> **Hyperparameters used in experiments**
>
> Apologise for not including the learning rate in the empirical experiments and will do so in the camera ready. Note that our synthetic model does however explicitly consider the effect of the learning rate. As shown in Fig.3b, discussed in sec. 4.2 and expanded upon in Appendix E.1 of the paper, the learning rate can act as a source of bias by advantaging the cluster with smaller variance.
>
> Following your suggestion, we tested the ‘standard configuration’ with varying *learning rate* (Figure 5 of the included PDF). As predicted by the theory (see sec. 4.2 and Figure 3b of the paper), for high learning rates, the loss of the higher variance group blows up (left panel of Figure 5 of the included PDF). For smaller learning rates, the dynamics remain the same, it is simply the overall speed that changes as we can see in the included figure (for smaller learning rates, it takes longer for the crossings to occur).
>
> *Weight decay* is quite an interesting technique to consider and our analysis offers direct predictions based on the theory. As discussed in Appendix E, smaller student magnitude advantages the higher variance group and indeed when we empirically include a weight decay penalty of 0.1 (Figure 4 right panel of the included PDF), we observe that asymptotically the higher variance cluster is now preferred since the penalty encourages smaller weights.
>
> As for *Adam*, we see very similar behaviour (Figure 4 centre panel of the included PDF) although we had to use a slower learning rate of 0.001 to prevent losses from blowing up since Adam leads to faster optimization manifesting in a higher effective learning rate than SGD. We fully expect all our key findings to translate.
>
> **Suggested additional experiment**
>
> This is a great suggestion. We provide results (see submitted PDF) from additional experiments to show how modifying the real data changes behaviour exactly as predicted by the theory. Figure 1 shows an extension of the CIFAR10 experiment discussed in the paper where we now vary the relative representation of the groups (darker colours reflect greater imbalance) and we see that as predicted by our theory, crossings only occur when the higher variance group (red) has lower relative representation. Notice that the dynamics of the initial phase seem almost unaffected by the relative representation and are determined only by the variance as predicted by the theory.
>
> We also extend the MNIST experiment discussed in the paper and show how increasing variance of the second group (red), controlled through image brightness, leads to faster learning in the second phase in Figure 3.
>
>
> **In conclusion**, we hope these extensions provide strong evidence showing how the factors identified in the theory affect real data experiments exactly as predicted.
>
> We also thank the reviewer for pointing out additional references that we will include in the further work.
> If there are any remaining concerns or if further clarification is needed, we would be happy to engage in additional discussion to address them.

---

> ### Author Response · Authors · 2024-08-13
>
> We hope that the clarifications and additional experiments have addressed the reviewer’s  concerns. If there are any remaining questions or points that need further discussion, we would be grateful for any additional feedback.

---

### Official Review · Reviewer_gHK2 · 2024-07-18

**Soundness:** 4
**Presentation:** 4
**Contribution:** 3
**Rating:** 6
**Confidence:** 3

**Summary:**

The paper investigates the emergence of bias in the transient phase of learning by gradient descent (after initialization and before convergence). The authors theoretically study a mixture of Gaussians dataset in a teacher-student framework. The paper then analytically characterizes several properties of a linear model applied to this dataset, showing the emergence of different forms of bias at different timescales. The authors then demonstrate these theoretical predictions empirically on rotated MNIST and CIFAR10 tasks.

**Strengths:**

**Originality**
The paper is original; although there are many prior works studying bias and fairness from a theoretical perspective, this paper studies these phenomena in the transient learning regime, which is uncommon. Moreover, the authors select a theoretical setting which can be characterized quite precisely.

**Quality**
The quality of the theory in the paper is strong. The theorems appear correct and support the overall claims of the paper. Experiments also seem to be generally conducted well.


**Clarity**
The paper is generally well-written. The authors do a particularly good job of illustrating figures, with good color choices and error bars when appropriate.


**Significance**
The paper may have some value to researchers studying bias and fairness from a theoretical angle. Certainly, this paper suggests that the theoretical setting used by the authors and the analytical techniques they use are worth pursuing further.

**Weaknesses:**

In my view, the main weakness of the paper is that its practical value and significance remains difficult to understand. The authors show that their theoretical predictions line up with empirically observed double crossing phenomena, which is great. However, the significance of this finding is not stated. I would encourage the authors to more explicitly draw a connection between their findings and practical problems. Bias and fairness are areas with major significance in the field, so I am reasonably confident that the papers' contributions will have some practical value, but this needs to be explicitly stated and discussed.

Minor comments:
- Typo on line 314

**Questions:**

What are the practical insights that can be drawn from this work? Can the authors elaborate more on what bias-mitigating mechanisms might arise from the paper's observations?

**Limitations:**

Limitations are adequately addressed.

---

> ### Author Rebuttal · Authors · 2024-08-07
>
> We thank the reviewer for their insightful comments and careful consideration of our paper.
>
> **On the practical significance of our work**
>
> Thank you for highlighting this important point. We plan to address the practical implications of our findings by adding the following discussions to the revised manuscript:
>
> * **Caution Against Implicit Assumptions:** Our results caution against the common implicit assumption in the literature on spurious correlation mitigation that bias remains the same throughout training (e.g., [1, 2]).
>
> * **Identifying Key Drivers of Bias:** Connecting our theoretical insights to actionable practices,  we identify variance–specifically image brightness–as a key factor influencing bias in neural networks, as discussed in our CIFAR 10 experiment (section 5). This finding enriches the literature on simplicity bias by providing a direct explanation for what might make a subpopulation appear ‘easier’ for a model to learn, thereby introducing bias.
>
> * **Expanding Bias Mitigation Techniques:** Most bias mitigation strategies focus on under/oversampling to adjust the relative representation of groups. Our work suggests that e.g., by normalising image data to have similar brightness across subpopulations,  practitioners can address an additional source of bias–group variance. This expands the toolkit available for developing bias-free bias-free classifiers, offering a more nuanced approach to bias mitigation.
>
> We hope these additions will clarify the practical significance of our work, and we are open to further discussion to ensure your concerns are fully addressed.
>
> **References**
>
> [1] E. Z Liu, B. Haghgoo, A. S. Chen, A. Raghunathan, P. W. Koh, S. Sagawa, P.  Liang, C. Finn.  Just Train Twice: Improving Group Robustness without Training Group Information. ICML 2021.
>
> [2] Y. Yang, E. Gan, G. K. Dziugaite, B. Mirzasoleiman. Identifying Spurious Biases Early in Training through the Lens of Simplicity Bias. arXiv: 2305.19761.

---

> > ### Comment · Reviewer_gHK2 · 2024-08-12
> >
> > Thank you for your thoughtful reply! I'm convinced that the additions made by the authors will clarify the practical significance of the work, and maintain my rating in favor of acceptance.
> >
> > The main weakness in my view is that (as echoed by some other reviewers), the theoretical setting is limited, which limits the impact of the work primarily to theoreticians at this stage (and practical experiments would be outside the scope of this paper). Nevertheless, the theoretical analysis is strong, which is why I am in favor of acceptance.

---

### Official Review · Reviewer_JTC4 · 2024-07-18

**Soundness:** 3
**Presentation:** 3
**Contribution:** 3
**Rating:** 5
**Confidence:** 4

**Summary:**

In this paper, the authors explore how bias evolves while learning features across different sub-populations. They particularly emphasize the transient phase of learning, which is less understood compared to the early and late phases of the learning dynamics.  To this end, data from various sub-populations are modelled with a Gaussian-mixture model, and this data is labeled by a teacher network.  A linear student it trained with online SGD and its dynamics are studied at a high-dimensional limit. Using closed-form analytic expressions at this limit, the bias evolution is studied in various settings, revealing interesting dynamics that are experimentally validated on real datasets.

**Strengths:**

a) The paper proposes a theoretical framework to study the class imbalance beyond the asymptotic regime and interesting training dynamics have been analysed.

b) The description of various timescales where generalization error of sub-populations dominated and the analytical characterisation of relevant quantities controlling the timescales is novel.

c) The paper is well written and the main arguments are easier to follow.

**Weaknesses:**

a) The paper studies only a linear student and it does not accurately capture the current over-parameterised deep learning models. This curtails the significance of the work.

b) One other weakness is training with online SGD on the whole population loss in comparison of the empirical loss, hence these dynamics might fall short of capturing the real-world training setup. It is also not clear if these closed form analytic expressions are tractable beyond $m=2$.

**Questions:**

a) Did the authors try any empirics with over-parameterized students over the Gaussian mixture, does it have any qualitatively different behaviour?

**Limitations:**

The limitations are adequately address in conclusion.

---

> ### Author Rebuttal · Authors · 2024-08-07
>
> We thank the reviewer for their appreciation of our characterisation of the non-asymptotics training dynamics and their constructive comments.
>
>
> **Regarding the Use of Linear Classifiers:**
>
>  As the reviewer points out, our theoretical analysis is limited to linear classifiers to ensure analytical tractability and to obtain,  for the first time to our knowledge,  closed-form expressions for the generalisation error curves and order parameters.
>
> However, we have also tested the robustness of our theory on more complex architectures. In Section 5 and Appendix F, we provided experiments using 2-layer MLPs and fine-tuned a ResNet-18 on real data sets. Taking your remark and the related remark of reviewer CSav into account, we have now included additional results that demonstrate the behaviour predicted by our theory on deeper and wider MLPs (see the submitted pdf in our global response):
>
> - We have expanded the experiment in section F.2 as follows: We consider a data distribution with $d=100$, $\rho=0.7$, $\Delta_+=0.1$, $\Delta_-=1$, $T_{\pm}=0.9$, $v=4$, $\alpha_+ = 0.471$,  $\alpha_- = -0.188$. We train a 2-layer MLP with ReLU activation in the hidden layer(s) and sigmoidal activation at the output with 128 neurons in the hidden layer with online SGD with a learning rate of 0.01 using BCE loss. We refer to this as the ‘standard configuration’ useful for further comparisons. We visualise the loss curves and test error rates on the positive (blue)  and negative (red) subpopulations along with standard deviation across 10 seeds in Figure 4 (left panel).
>
> - We further train MLPs of increasing depth (2,3,4,5 layers) with 128 hidden neurons in each layer and visualise the loss and error curves in Figure 6. The figures show the three phase behaviour predicted by our theory - the positive subpopulation (in blue) is learnt faster initially since it exhibits stronger spurious correlation with the shift vector; the negative subpopulation (in red) is learnt faster subsequently due to higher variance and finally the positive subpopulation is asymptotically prioritised once more due to higher relative representation. As such, we also observe the ‘double-crossing’ phenomena discussed in the paper. We stress that this result and the next are obtained by taking exactly the same parameters used in Fig.10 of the paper without exploring the phase space any further.
>
> - We also train MLPs of 2 layers but increasing width (2, 16, 128 and 1024 hidden neurons) and visualise loss and errors curves in Figure 7. We note the presence of the same three phase behaviour across increasing parameterization. Thus, our empirics with over-parameterized students show qualitatively similar behaviour. We hope this further empirical validation helps show the validity of our theory in settings closer to modern regimes.
>
> **Regarding Online SGD with Empirical Loss:**
>
>  We would like to clarify that our theory indeed considers with online SGD using the empirical loss. Specifically, at each time step, we sample a new supervised example, calculate the loss for that example, and then update the model parameters using gradient descent. All our empirical experiments also follow this training procedure.
>
> While this approach is standard in theoretical analyses due to its analytical convenience, we acknowledge the importance of capturing the dynamics of multi-pass/batch SGD, as we discussed in the 2nd paragraph of our conclusion. In future work, we aim to address this challenge by leveraging results from dynamical mean field theory (e.g., [1]).
>
> **On tractability beyond m=2**:
>
> Our current work focuses on $m=2$ to facilitate the  interpretation of the dynamics and to identify key takeaways. However, we are confident that the equations remain analytically tractable for higher $m$ values. Adding more subpopulations would involve taking weighted averages by representation in all expectations,  similar to the existing analysis. We anticipate that this lead to a system of coupled ODEs similar to what we presented in Appendix D.1.
>
> If there are any remaining concerns or if further clarification is needed, we would be happy to engage in additional discussion to address them.
>
> **References**:
>
>   [1]  F. Mignacco, F. Krzakala, P. Urbani, L. Zdeborová. Dynamical mean-field theory for stochastic gradient in Gaussian mixture classification. NeuriPS 2020.

---

> > ### Comment · Reviewer_JTC4 · 2024-08-12
> > **reply to the author's rebuttal**
> >
> > Thank you for conducting additional experiments on over-parameterization. The experiments provide some clarity on the relevance of the phenomenon beyond linear classifiers. However, I have decided to keep my score since the theoretical analysis of the phenomenon is limited to linear setting.

---

### Author Rebuttal · Authors · 2024-08-07

We thank all reviewers for their insightful comments and careful consideration of our paper. Their feedback  has helped us identify ways to improve our presentation and convey our message more clearly. It has also prompted us to conduct additional experiments,  which we hope demonstrate the robustness of our theory and provide clearer explanations of our results.

We have included a supplementary PDF with additional figures and have explained in detail their relevance in individual responses to the reviewers. However, for the benefit of everyone’s understanding,  we would like to briefly highlight some of the key experiments here.

* **Figure 1**: This figure presents an extended version of the CIFAR10 experiment,  where we systematically vary the relative representation of the groups (darker colours represent greater imbalance against the red group). As predicted by our theory, relative representation primarily affects the later dynamics, while the middle dynamics are governed only by group variance.

* **Figure 2**: We show the learning dynamics on MNIST of a simplified architecture with respect to the one considered in section 5 of the paper where the hidden layer doesn't have non-linearities. The result shows consistent behaviour with Figure 5 of the main text.

* **Figure 3**: Here, we extend the MNIST experiment by varying the average image brightness of the second group, which reflects the group variance in our theory. The results show that greater brightness leads to faster learning in the second phase and an increasing asymptotic preference, consistent with our predictions.

* **Figures 4, 6, 7**: In these figures, we extend the experiment in Appendix F.2,  where an MLP with non-linearities is trained on synthetic data  with BCE loss. Across different optimizers and varying degrees of parameterization, we observe a three-phase behaviour:  the positive subpopulation (in blue) is learned faster initially due to a stronger spurious correlation with the shift vector; the negative subpopulation (in red) is learned faster subsequently due to higher variance,  and finally, the positive subpopulation is asymptotically prioritised once more due to higher relative representation.

We believe these additional results strengthen the contributions of our paper and we hope they address the reviewers’ concerns. We welcome any additional questions or remarks and sincerely appreciate the reviewers’ time and effort throughout this review process.

The authors

---

### Author Response · Authors · 2024-08-14
**Authors-Reviewers Discussion Summary**

As the author-reviewer discussion period comes to a close, we would like to extend our sincere thanks to the reviewers and the area chair for their time and thoughtful feedback.

## New Results

In response to the reviewers' suggestions, **we have conducted an extensive series of additional experiments across a variety of datasets and network architectures**, demonstrating the robustness of our theoretical findings beyond the simplified settings initially considered. As shown in the supplementary experiments included in our rebuttal, our empirical results align remarkably well with our theoretical predictions, providing valuable insights into practical applications. Notably, our experiments reveal robustness across different optimization techniques, including variations in learning rates, Adam optimizer, and weight decay.

## Contributions

A key concern raised by the reviewers was that our theoretical analysis focuses on linear classifiers, which differ significantly from modern deep-learning architectures. We would like to emphasize that most theoretical studies on spurious correlations and data imbalance still employ linear classifiers due to the complexity of these problems [1,2,3,4,5]. **Within the current theoretical landscape**, our paper makes a **significant technical contribution** by providing, for the first time, a closed-form solution of the dynamics and **predicting changes in bias** during training. These insights have practical implications, particularly for debiasing algorithms [6,7], as discussed in our response to Reviewer gHK2.

Moreover, our study reveals that even for linear classifiers, common assumptions can be misleading. Our simulations demonstrate robustness beyond what is captured by theoretical analysis. Empirically, the trend observed [4,8] suggests that increased model complexity tends to amplify bias.

## Conclusion

While we regret that we were unable to engage with all reviewers during the discussion period, we remain hopeful that the forthcoming discussions will lead to a favorable outcome.

Best regards,
The Authors

## References:

[1] Loffredo, E., Pastore, M., Cocco, S., & Monasson, R. (2024). Restoring balance: principled under/oversampling of data for optimal classification. In ICML 2024.

[2] Hermann, K. L., Mobahi, H., Fel, T., & Mozer, M. C. (2024). On the Foundations of Shortcut Learning. ICLR 2024.

[3] Nagarajan, V., Andreassen, A., & Neyshabur, B. (2021). Understanding the failure modes of out-of-distribution generalization. ICLR 2021.

[4] Sagawa, S., Raghunathan, A., Koh, P. W., & Liang, P. (2020). An investigation of why overparameterization exacerbates spurious correlations. In ICML (pp. 8346-8356). PMLR.

[5] Takahashi, T. (2024). A replica analysis of under-bagging. TMLR.

[6] Liu, E. Z., Haghgoo, B., Chen, A. S., Raghunathan, A., Koh, P. W., Sagawa, S., ... & Finn, C. (2021). Just train twice: Improving group robustness without training group information. ICML 2021 (pp. 6781-6792). PMLR.

[7] Yang, Y., Gan, E., Dziugaite, G. K., & Mirzasoleiman, B. (2024). Identifying spurious biases early in training through the lens of simplicity bias. In AISTATS (pp. 2953-2961). PMLR.

[8] Bell, S. J., & Sagun, L. (2023). Simplicity bias leads to amplified performance disparities. In Proceedings of FAccT 2023 (pp. 355-369).

---

### Decision · Program_Chairs · 2024-09-25

**Decision:**

Accept (poster)

**Comment:**

The paper analyzes the training dynamics (with ODEs) of a linear model trained with SGD under an assumed Gaussian mixture data distribution. Notably the paper makes precise how the optimization dynamics affect data subgroups in this specific setup. Among other settings, the presented theoretical results are particularly relevant for learning under fairness considerations, and learning in the presence of spurious features. While other existing works tend to rely on asymptotic analysis, the present work’s goal is to understand  the complex time-dependent nature of bias during training.

**Review**

All reviewers' comments were positive on the strength of the theoretical analysis in the specific setting considered. Notably, the closed-form time-dependent characterization of various quantities (JTC4), and the prediction of the double-cross phenomenon (CSav) are novel. Due to the assumed Gaussian mixture model, the issues of practical value and significance came up during the discussion. The authors sufficiently addressed these points by providing more experiments on more datasets and network architectures. Another issue raised was that the paper only considers a linear model. On this point, the AC thinks that the value of the contribution is in its full characterization (closed-form solution) of training dynamics under a distribution-model setup, even though that setup may be quite specific. The AC thinks that characterizing training dynamics in the more general case is not easy, and that this contribution can provide a stepping stone for a more complicated data distribution model that is yet to come in future works.

Recommendation: accept.